# Case Study on Life Cycle Assessment Applied to Road Restoration Methods

Gislaine Luvizão [1,*] and Glicério Trichês [2]

1   Department of Civil Engineering, University of West Santa Catarina, Joaçaba 89600-000, Brazil
2   Department of Civil Engineering, Federal University of Santa Catarina, Florianópolis 88040-900, Brazil
*   Correspondence: gislaine.luvizao@unoesc.edu.br

**Abstract:** Brazil's dependence on road transportation, combined with the high extent of the network and the lack of investment management in maintenance and restoration, makes traffic conditions poor, resulting in unwanted costs and environmental impacts. Life cycle assessments are a promising tool that assists in decision making. This study aimed to evaluate the environmental performance of three roads, applying different restoration and maintenance techniques throughout the analysis cycle. To develop this study, the ecoinvent database and the OpenLCA software were used to model, based on studies developed in the HDM-4 (Highway Development and Management) software, and the interventions were applied for the initial year and for 30 years. Using the life cycle assessment methodology, the environmental impacts generated for the categories of acidification, climate change, eutrophication, ecotoxicity, human toxicity and photochemical oxidation were identified. The results show that, when analyzing the restorations in their implementation, deep recycling generates more environmental impacts; however, when planning the restorations throughout the cycle, deep recycling becomes, on average, 47% less impactful than the structural reinforcement technique, which is the same behavior that has been identified with the Whitetopping technique. It becomes evident that the use of rigid structures, such as Whitetopping or semi-rigid structures and deep recycling with Portland cement additions, generate fewer environmental impacts when compared to flexible structures that consume a large amount of asphalt binder, and that require interventions at shorter intervals.

**Keywords:** life cycle assessment; environmental impact; restoration; roads; Whitetopping; reinforcement; recycling; HiMA

## 1. Introduction

The economic, political and social development of any country is linked to its infrastructure, especially transportation. In Brazil, only 12.4% of the road network is paved, and the country relies on the road sector for passenger (95%) and cargo (61%) transportation [1].

In 2013, road transportation was responsible for 23% of $CO_2$ emissions in the 28 countries of the European Union [2]. Federal public investments in roads are low, and a more efficient management system is needed. In addition to assessments based on structural performance and costs, environmental impacts must also be part of the decision-making process for restoration and maintenance strategies [3].

The management system aims for sustainable development through a balance among the environment, society and the economy, meeting current needs without compromising the future and improving the political, economic, social, production, technological, international and administrative systems of the government and industries [4,5]. The use of environmental approaches for decision making in construction projects is becoming more common, implemented through waste management actions, through the control of the exploitation of natural resources and through reductions in noise, gaseous, liquid or solid pollution [3].

The life cycle assessment (LCA) methodology is a useful tool for achieving a complete overview of a product or process. LCA is an essential process for achieving conclusions and maintenance strategies or global projects for the entire lifespan, standardizing functional units, expanding the scope of studies, improving quality and reliability and expanding boundary systems [6].

The reliability and accuracy of an LCA are affected by the reliability of the adopted methodologies and models. Moreover, these models require the estimation of input parameters, inventory development and methodological choices, which may impact the results significantly [7].

$CO_2$ emissions are the main contributors to global warming, and often, a simple inventory of their emission levels is sufficient as an impact indicator [6]. The main environmental impacts studied in LCA are global warming (climate change), ozone depletion, human toxicity, ionizing radiation, sensory disturbances, photo-oxidation, acidification, eutrophication, ecotoxicity, land use and resource depletion [8,9].

The construction sector is responsible for substantial consumption of energy and natural resources; thus, the construction and maintenance of a road system has a significant impact on the environment [10,11]. Because maintenance expenses typically comprise half of the annual road infrastructure funds, it is very important to prioritize efficiency in road maintenance [12]. Most of the road investment is related to the exploration of raw materials, the labor for execution and the transport of materials, which comprise most of the maintenance and operation costs [13]. In Brazil, approximately 12.5% of the total budget invested in São Paulo, in 4 years, was applied in the development and maintenance of roads [14].

In order to choose the process with the least impact, several environmental impact indicators must be considered, because the contribution of each step related to the LCA differs significantly between each analyzed parameter. High volumes of traffic combined with the geometric layout present variations in the degree of environmental sustainability. Therefore, social and economic criteria must be integrated in the comprehensive evaluation of road works [15]. Considering the LCA from construction to use, it was identified that roads that do not need tunnels and bridges have lower environmental impacts. Flexible pavements present lower initial impacts when compared to rigid ones; however, when considering prolonged periods of analysis, the behavior is reversed [16].

Various studies are being carried out in the area of environmental impacts. Although some are presented as LCAs, they only contain life cycle inventories (LCI) and do not provide a life cycle impact assessment (LCIA) [6]. However, there are notable differences in all LCA studies of pavement resulting from variations in the included activities, which is a factor that is often due to resource restrictions. Meticulous quantification of the environmental impacts of pavements requires information from numerous sources related to the stages of their life cycle, which is information that is not always available. LCA studies are subject to assumptions and simplifications regarding their scope, system limits and data, inevitably leading to uncertainties in LCA assessments [17].

The lack of standardization in LCIAs has been discussed against temporal standards, in relation to the use phase with a study on carbonation, albedo [18] and roughness [19]; in relation to dynamic standards eutrophication [20], human toxicity [21] or the impact on human health from noise [22]; in relation to recycled asphalt (RAP) [23–32]; in relation to recycled concrete pavement (RCP) or recycled asphalt shingles (RAS) [25,28,30,33,34]; in relation to the use of fly ashes [28–30,32,33,35]; in relation to the development of the pavement construction and maintenance model [29]; in relation to the evaluation of warm asphalt mixtures [36]; in relation to the analysis of concrete pavement with an industrial byproduct [37]; in relation to the quantification of the environmental benefits of in situ recycling [38]; and in relation to the analysis of the rolling resistance of pavement and greenhouse gas emissions [39,40]. In Europe, EN 15804:2012+A2:2019 is used to conduct environmental studies through the life cycle assessment of construction products and services; however, it does not include social and economic studies [41].

Asphalt-coated pavement is 44% less impactful than concrete pavement in terms of environmental performance. The determining factor for the low performance of concrete pavement is mainly due to $CO_2$ emissions related to cement manufacturing and its contribution to global warming [42]. However, in other studies, concrete pavement produced 40% more $CO_2$ emissions than those produced by asphalt pavement [43]. Portland cement concrete consumes more energy than that consumed by asphalt pavement [44]. When analyzing three different pavement rehabilitation options, it was found that energy consumption is higher for the asphalt option, whereas the impact on global warming is higher for the Portland cement concrete option [45].

When comparing different bituminous mixtures containing recycled materials, namely crumb rubber (CR) and recovered asphalt pavements (RAP), through the results of a life cycle assessment (LCA), it was identified that the use of CR in the production of asphalt mixtures shows a reduction in the need for the gross energy ratio (GER) and greenhouse gas (GHG) emissions [46,47]; a reduction in noise, especially when the dry process is used [48]; and improvements in cushioning properties when CR is mixed as an aggregate as a result of vibration absorption by rubber [49].

The energy of the raw material in the asphalt binder was reported as being 40.2 MJ/kg [6]. According to the Energy Research Company [50], 78.2% of Brazil's electrical matrix is renewable, and in the world, this value is 22.2%.

The amount of GHG emissions induced at a vehicle speed of 32 km/h can be doubled with an increase in the International Roughness Index (IRI) from 1 to 8 m/km [51]. The presence of a work zone, resulting from the closure of lanes, roads and detours during construction and maintenance events, affects traffic flow, producing delayed traffic, a congestion impact on the road network and an overall increase in vehicle fuel consumption [7]. For equipment using diesel fuel, the GHG emissions values are 79 g/MJ for $CO_2$, 0.0016 g/MJ for $N_2O$ and 0.00005 g/MJ for $CH_4$ [52]. The energy consumption by the traffic acting during the life of a road is of utmost importance, being responsible for about 95 to 98% of the total energy consumption imputable to it, with the construction, maintenance and operation activities responsible for the remaining 2 to 5% [53].

The analysis period (life cycle) affects the results and policies adopted for the maintenance of the works. In Germany, for example, secondary waste was studied as a substitute for primary materials over a period of 100 years [54]; in China, concrete pavements with a life cycle of 20 and 40 years were studied [55]; in Korea, different civil structures were compared over 50 years [56]; and in Spain, sustainable maintenance program methods of pavements under budget constraints were analyzed, considering periods of 25 years. However, there are studies that have been developed that utilize analyses for between 15 and 50 years [57]. In the United States of America, one of the countries that develops the most studies in the LCA area, the life cycle of asphalt pavement recycling was evaluated with respect to GHG emissions with a temporal aspect of 40 and 100 years [58].

This study aims to combine the use of the HDM-4 software with the annual analysis of maintenance and restorations together with studies of the environmental impacts that have been obtained in life cycle assessments (LCAs), contributing technically and scientifically to decision making in choosing the appropriate alternative. It provides the necessary environmental parameters for the good performance of road projects, taking into consideration the life cycle of materials and structures, reducing pollutant emissions and improving the quality of life for users over a 30-year period.

First, case studies are presented with their respective characteristics, including the results obtained in the HDM-4 software with the frequency of interventions, followed by the quantification of consumption for all stages, from raw material extraction to the end of the analysis cycle, obtaining the inventory. The OpenLCA [59] software is used for environmental modeling, followed by the presentation and analysis of the results of the environmental impacts, generated by each type of intervention adopted for each impact category, providing parameters to support decision making in the management of roads.

## 2. Materials and Methods

To fill the bibliographic gap regarding environmental studies in 30-year life cycles, with a focus on decision making for the choice of road restorations and maintenance applied to traffic and road structure characteristics, the characteristics of three different roads located in the state of Santa Catarina, Brazil, were evaluated. These constructions were chosen because they present the structures and methodologies of restorations that are targeted by this study, and because we have knowledge of the characteristics of the materials (laboratory tests), the executive processes that were used and evaluations of post construction performance.

In order to simplify the identification of each case study, the denomination "SC XXX Y/Z" is adopted, with SC representing a road in the State of Santa Catarina, XXX being the classification/name of each road (355 and 114), Y being the city that is considered the origin of the segment (J–Jaborá, P–Painel, L–Lages) and Z being the destination of each segment (BR 153, SJ–São Joaquim, OC–Otacílio Costa). Therefore, the following denominations are used: SC 355 J/BR153, SC 114 P/SJ and SC 114 L/OC.

For the three roads, the restoration dimensioning methods PRO 11 (DNER, 1979), PRO 269 (DNER, 1994) and Deep Recycling (using a case study of a built and monitored work on SC303–current SC150) were used. For SC114 L/OC, in addition to the mentioned methods, PCA/84 was considered for rigid pavements, using Whitetopping. In the segment between Painel and São Joaquim (SC114 P/SJ), the use of the highly modified asphalt mixture HiMA, as a replacement for the conventional binder, was simulated using the ME-Design software.

Due to the fact that the executive projects consider 10 or 20 year life cycles for the adopted restorations, the HDM-4 software was used for the simulation of the maintenance and restoration possibilities, fed with the data obtained in the design phase regarding cracked areas, deflection, LII, traffic and climate. Table 1 presents the typical sections that are considered in each studied road.

**Table 1.** Typical sections of the studied roads.

| Nome | SC355 J/BR153 | SC114 P/SJ | SC114 L/OC |
|---|---|---|---|
| Highway Class | Secondary | Secondary | Secondary |
| Length (km) | 1 | 1 | 1 |
| Lane Width (m) | 6.6 | 7.0 | 7.0 |
| Shoulder Width (m) | 2.4 (2 × 1.20) | 3 (2 × 1.50) | 2.4 (2 × 1.20) |
| Traffic Direction | Two-way | Two-way | Two-way |
| Surface Class | Asphalt | Asphalt | Asphalt |

The traffic was classified according to the number of vehicles that travel on the road under study. The SC355 J/BR153 and SC114 P/SJ highways were classified as medium traffic, and SC114 L/OC was classified as high traffic (Table 2). The volume of traffic directly impacts the restoration technique adopted, the maintenance periodicity and the costs to ensure minimum safety and comfort for users.

**Table 2.** Classification of traffic according to the number of vehicles that travel on the road.

| SC | Traffic Range | Average Daily Volume (Vehicles) | |
|---|---|---|---|
| | | Starting Age | Final Age |
| 355 J/BR153 | Medium | 2538 | 5721 |
| 114–P/SJ | Medium | 2164 | 5489 |
| 114–L/OC | High | 5758 | 13,814 |

After feeding the HDM-4 software with all configuration, calibration and current road condition data, techniques were adopted to apply restoration and maintenance, all in accordance with the executive project of each road. Although the resurfacing methodology is the same for the three works, different asphalt mixture thicknesses were adopted due

to the fact that the roads have different traffic and structural conditions. Figure 1 presents the organization of the case study analysis in the HDM-4 software. The evaluations are separated by road, adopting the maintenance related to each type of restoration, making it possible to finally elaborate the life cycle inventory for each simulation.

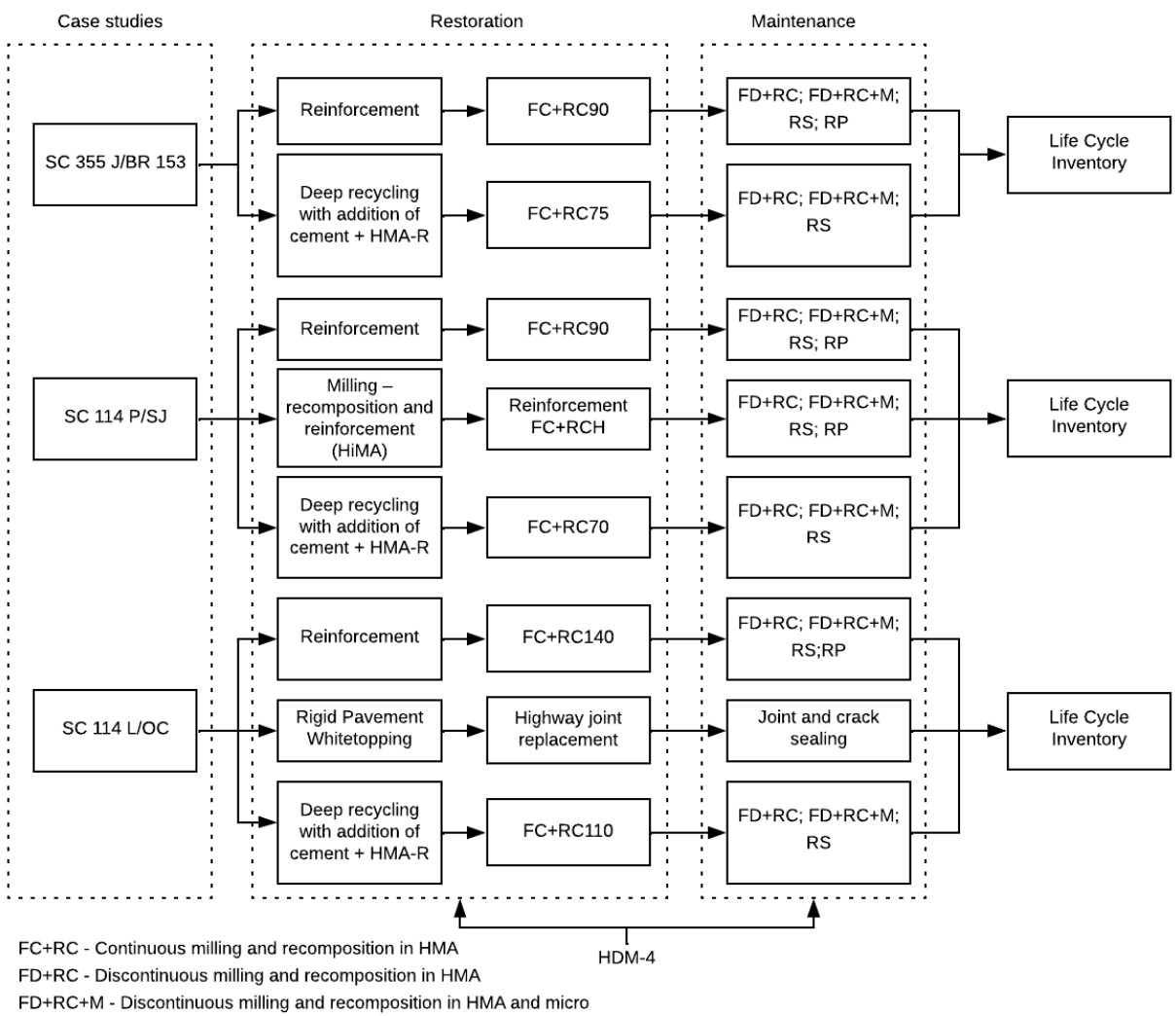

**Figure 1.** Organization of the case studies in the simulations in the HDM-4 software.

Therefore, the initial methodology for restoring the pavements was adopted as follows:

(a) SC355 J/BR153: (1) Structural reinforcement with 10 cm thickness; (2) Deep recycling with cement addition (18 cm) and an overlay layer (3 cm SAMI plus 5 cm HMA rubber).

(b) SC114 P/SJ: (1) Structural reinforcement with 5 cm thickness; (2) Milling of 4 cm of the existing layer with recomposition (2 cm SAMI layer plus 4 cm HMA HiMA layer); (3) Deep recycling with cement addition (18.5 cm) complemented with an overlay layer (3 cm SAMI plus 5 cm HMA rubber).

(c) SC114 L/OC: (1) Structural reinforcement with 15 cm thickness; (2) Deep recycling with cement addition (20 cm) and an overlay layer (4 cm SAMI plus 8 cm HMA rubber); (3) Rigid pavement (Whitetopping—concrete plates with 22 cm).

Throughout the 30 years, new large-scale restoration and maintenance interventions were necessary. As a result, each approach in each case study had the following configuration:

(a)     SC355 J/BR153

Reinforcement option: Begins in 2014 with structural reinforcement with HMA with a 10 cm thickness; in 2019, maintenance through FD+RC+M; in 2023, restoration with FC+R90; in 2028, FD+RC; in 2031, new FC+R90; in 2036, execution of FD+RC+M; in 2039, the last restoration with FC+R90.

Recycling option: Begins in 2014 with the execution of deep recycling with cement addition; in 2021 maintenance with FD+RC technique; in 2025 necessary new restoration with FC+RC75; in 2031 maintenance through FD+RC+M; in 2037 with restoration in FC+RC75.

(b)     SC114 P/SJ

Reinforcement option: Begins in 2014 with the structural reinforcement in HMA with 5 cm thickness; in 2018 maintenance through FD+RC; in 2021 restoration with FC+RC90; in 2026 FD+RC+M; in 2029 new FC+RC90; in 2034 execution of FD+RC; in 2037 the last restoration with FC+R90; in 2042 with FD+RC+M.

Recycling option: Begins in 2014 with the execution of deep recycling with cement addition; in 2021, maintenance with the FD+RC technique; in 2025, necessary new restoration with FC+RC70; in 2031, maintenance through FD+RC+M; in 2037, restoration with FC+RC70.

HiMA option: Begins in 2014 with continuous milling with a 4 cm thickness with the recomposition of two layers (2 cm SAMI plus a 4 cm asphalt mixture with HiMA); in 2019, maintenance with the FD+RC technique; in 2024, restoration through FC+RCH; in 2028, execution of maintenance with FD+RC+M; in 2032, execution of reinforcement restoration with a 5 cm thickness; in 2037, last maintenance with the execution of FD+RC; in 2040, FC+RCH.

(c)     SC114 L/OC

Reinforcement option:  Begins in 2017 with structural reinforcement with HMA with a 15 cm thickness; in 2020, maintenance through FD+RC; in 2023, restoration with FC+RC140; in 2026, FD+RC+M; in 2029, new FC+RC140; in 2032, execution of FD+RC; in 2034, restoration with FC+RC140; in 2037, maintenance with FD+RC+M; in 2039, FC+RC140; in 2042, FD+RC; in 2044, FC+RC140.

Recycling option: Begins in 2017 with the execution of deep recycling with cement addition; in 2022, maintenance with the FD+RC technique; in 2026, a necessary new restoration with FC+RC110; in 2030, maintenance through FD+RC+M; in 2034 and 2039, application of FC+RC110; in 2042, maintenance with FD+RC; in 2046, restoration with FC+RC110.

Whitetopping option: Begins in 2017 with the filling of shoulders and wheel tracks with an asphalt mixture and the execution of the concrete layer (Whitetopping); in 2024, maintenance through crack sealing and joint replacement; in 2027, partial joint replacement; in 2029 and 2034, new crack sealing and joint replacement; in 2037, partial joint replacement; in 2039 and 2044, new crack sealing and joint replacement.

*2.1. Material and Fuel Consumption*

The quantification of services related to the execution of the initial restoration (Table 3), restorations and maintenance of the pavements, which, as planned, will be carried out within the 30-year analysis period (Table 4), was obtained from the restoration and maintenance strategies proposed for the pavements by HDM-4 and the unit compositions contained in DNIT's Reference Cost System for Works–SICRO.

Table 3. Inventory for the execution of the initial intervention.

| Road | SC 355 J/BR153 | | SC 114 P/SJ | | | SC 114 L/OC | | |
|---|---|---|---|---|---|---|---|---|
| Initial Intervention | Recycling | Reinforcement | Recycling | Reinforcement | HiMA | Recycling | Reinforcement | Whitetopping |
| Diesel (L) | 34,800.8 | 27,066.4 | 33,642.8 | 17,485.9 | 19,641.5 | 41,080.3 | 38,786.9 | 28,973.8 |
| Energy (kW) | 4006.8 | 4692.5 | 4308.1 | 3076.2 | 3057.0 | 8278.7 | 9698.3 | 5044.8 |
| Fuel oil (l) | 11,520.0 | 14,976.0 | 12,480.0 | 8448.0 | 8064.0 | 31,449.6 | 24,768.0 | 3763.2 |
| Asphalt mixture (t) | 213.8 | 483.1 | 126.0 | 294.0 | 168.0 | 126.0 | 294.0 | 361.0 |
| Petroleum asphalt emulsion (t) | 28.1 | 19.7 | 30.9 | 13.7 | 16.9 | 29.5 | 20.7 | 9.8 |
| Diluted petroleum asphalt (t) | 11.0 | 1.1 | 12.2 | 1.2 | 1.5 | 11.5 | 1.2 | 1.6 |
| Petroleum asphalt cement (t) | 79.2 | 103.0 | 85.8 | 58.1 | 55.4 | 123.6 | 175.2 | 26.6 |
| Portland cement (t) | 142.6 | 0.0 | 162.8 | 0.0 | 0.0 | 165.4 | 0.0 | 785.8 |
| Simple graded gravel (m$^3$) | 263.3 | 198.0 | 237.4 | 178.5 | 238.0 | 157.5 | 157.5 | 211.5 |
| Asphalt mix aggregate (m$^3$) | 567.0 | 1.163.1 | 614.3 | 656.1 | 626.3 | 1328.0 | 1830.3 | 278.1 |
| Macadam (m$^3$) | 237.0 | 178.2 | 265.3 | 199.5 | 266.0 | 210.0 | 210.0 | 282.0 |
| Soil (m$^3$) | 594.0 | 594.0 | 630.0 | 630.0 | 840.0 | 735.0 | 735.0 | 987.0 |
| Crusher run gravel (m$^3$) | 477.0 | 0.0 | 542.5 | 0.0 | 0.0 | 532.7 | 0.0 | 2504.3 |
| Lime (t) | 0.0 | 0.0 | 0.0 | 0.0 | 0.0 | 127.2 | 175.2 | 26.6 |
| Gasoline (L) | 0.0 | 0.0 | 0.0 | 0.0 | 0.0 | 0.0 | 40.0 | 727.4 |

Table 4. Inventory for construction, maintenance and restoration over a 30-year cycle.

| Road | SC 355 J/BR153 | | SC 114 P/SJ | | | SC 114 L/OC | | |
|---|---|---|---|---|---|---|---|---|
| Initial Intervention | Recycling | Reinforcement | Recycling | Reinforcement | HiMA | Recycling | Reinforcement | Whitetopping |
| Diesel (L) | 69,208.9 | 117,319.2 | 67,105.3 | 109,117.2 | 83,559.9 | 124,911.9 | 233,340.3 | 40,514.8 |
| Energy (kW) | 10,558.3 | 16,405.6 | 10,835.4 | 17,363.6 | 13,089.4 | 36,018.5 | 55,442.0 | 5221.5 |
| Fuel oil (l) | 32,428.8 | 52,358.4 | 33,312.0 | 48,768.0 | 35,328.0 | 93,273.6 | 123,552.0 | 3763.2 |
| Asphalt mixture (t) | 213.8 | 1338.5 | 126.0 | 1201.2 | 798.0 | 126.0 | 294.0 | 361.0 |
| Petroleum asphalt emulsion (t) | 64.0 | 87.7 | 68.9 | 86.6 | 87.9 | 116.6 | 149.5 | 9.8 |
| Diluted petroleum asphalt (t) | 11.0 | 5.4 | 12.2 | 5.8 | 5.8 | 11.5 | 15.0 | 1.6 |
| Petroleum asphalt cement (t) | 222.9 | 360.0 | 229.0 | 335.3 | 242.9 | 561.0 | 874.1 | 26.6 |
| Portland cement (t) | 142.6 | 0.0 | 162.8 | 0.0 | 0.0 | 165.4 | 0.0 | 785.8 |
| Simple graded gravel (m$^3$) | 263.3 | 990.0 | 237.4 | 892.5 | 892.5 | 157.5 | 1207.5 | 211.5 |
| Asphalt mix aggregate (m$^3$) | 2190.8 | 4066.3 | 2232.1 | 3787.4 | 2743.7 | 5896.8 | 9130.3 | 278.1 |
| Macadam (m$^3$) | 237.0 | 891.0 | 265.3 | 997.5 | 997.5 | 210.0 | 1610.0 | 282.0 |
| Soil (m$^3$) | 594.0 | 2970.0 | 630.0 | 3150.0 | 3150.0 | 735.0 | 5635.0 | 987.0 |
| Crusher run gravel (m$^3$) | 576.0 | 198.0 | 647.5 | 210.0 | 105.0 | 638.5 | 211.7 | 2504.3 |
| Lime (t) | 0.0 | 0.0 | 0.0 | 0.0 | 0.0 | 564.6 | 874.2 | 26.6 |
| Gasoline (L) | 0.0 | 0.0 | 0.0 | 0.0 | 0.0 | 0.0 | 40.0 | 727.4 |

### 2.2. Life Cycle Assessment

This study is based on the following justifications: (a) flexible pavements are the most commonly used in roadworks in Brazil; (b) it is known that currently dimensioned and constructed flexible pavements have durability (with comfort and safety) for around 10 years, and rigid pavements are dimensioned for a lifespan of 20 to 40 years; and (c) the environmental assessment of the components (raw materials) of the pavement and the execution process of the road structure are still not part of the decision-making process of the Brazilian public sector.

The proposed analysis system has the function of ensuring safety and comfort to road users for a period of 30 years. To do so, the extraction process of the raw material, initial restorations, subsequent maintenance to ensure minimum performance and restorations when structural intervention is necessary will be considered.

The functional unit was adopted as 1 (one) linear kilometer, with a single lane and with a shoulder, with the characteristics presented in Figure 2.

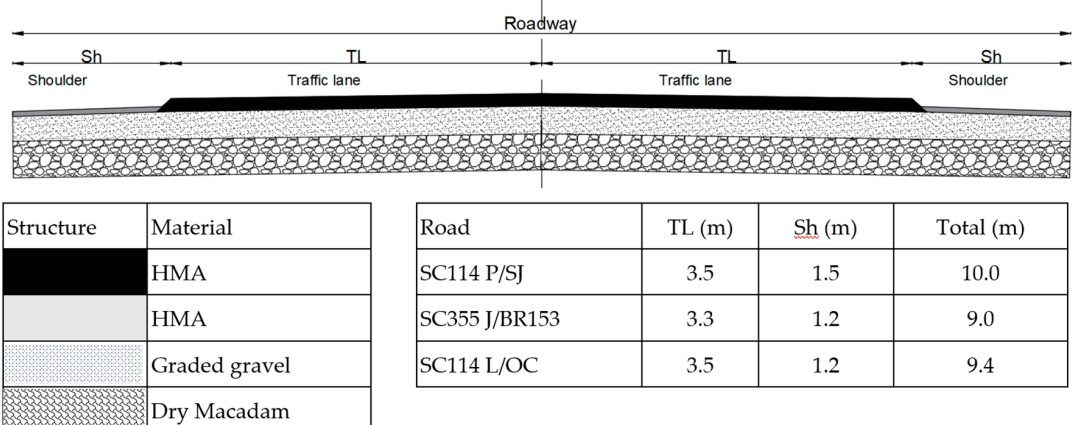

| Structure | Material |
|-----------|----------|
| | HMA |
| | HMA |
| | Graded gravel |
| | Dry Macadam |

| Road | TL (m) | Sh (m) | Total (m) |
|------|--------|--------|-----------|
| SC114 P/SJ | 3.5 | 1.5 | 10.0 |
| SC355 J/BR153 | 3.3 | 1.2 | 9.0 |
| SC114 L/OC | 3.5 | 1.2 | 9.4 |

Original data from experimental lanes.

**Figure 2.** Track geometry considered in the LCA for construction, restoration and maintenance.

Three stages were adopted in the life cycle of a pavement, which are presented below and detailed in Figure 3:

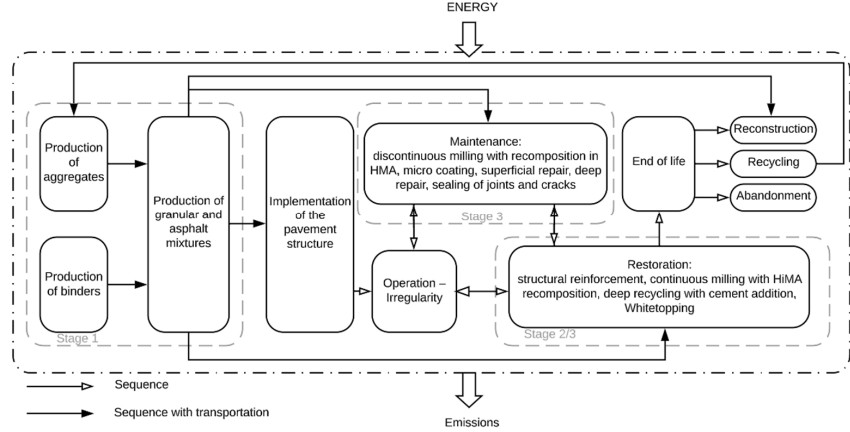

**Figure 3.** Steps considered in environmental modeling.

Stage 1—Production of raw materials and mixtures: In this stage, all inputs and all outputs of the system (inventory obtained from simulation in the HDM-4 software) are listed, including the extraction and crushing of aggregates, the production of binders (Portland cement, asphalt cement, cutback asphalt and asphalt emulsion) and the milling of mixtures (graded aggregate, asphalt mixtures and Portland cement concrete), in addition

to quantifying the transport of inputs from the extraction site to the place of production to the place of application, as well as the activities and equipment that are necessary for these services to be successfully completed.

Stage 2—Initial restoration execution: After approximately 30 years of implementation of the roads, they are restored, and this stage considers the activities that encompass the execution of the adopted initial restorations (deep recycling with cement addition, structural reinforcement, continuous milling with SAMI and HiMA and Whitetopping re-composition), considering the characteristics that the roads had in 2012. The transport and use of equipment that is necessary for the execution of the structure are covered.

Stage 3—Maintenance and restorations: Throughout the 30-year analysis, interventions are necessary to maintain comfort and safety for users; to do so, all raw materials that are necessary for the recompostion of the layers, the equipment and the transport that is used for execution are considered. The intervention periods are detailed in the executive process and modeled in the HDM-4 software. It does not include the initial restorations of the life cycle, only those that are necessary after the first intervention, in addition to ensuring the quality of trafficability with safety and comfort at the end of the 30-year analysis.

The boundaries were defined via a cutting process, i.e., via the delimitation of the phases, services and materials that were or were not considered.

(a)  Rock materials: The process is defined from the blasting of the rock, followed by transportation to the quarry, crushing and transportation to the application site or to the mixing plant (in kg).

(b)  Bituminous materials: The asphaltic binders (petroleum asphalt cement, petroleum emulsion and diluted petroleum asphalt, quantified in kg) are used in asphaltic mixtures, bond coating, primer and surface treatment, considering the exploration of petroleum, petroleum refining and transportation to the place of use (plant or track).

(c)  Portland cement: Used in the execution of concrete pavement (Whitetopping) and in deep recycling with cement addition, this material is modeled considering clinker production (in kg), cement production (in kg) and transportation to the place of use (plant or track).

(d)  Soil: In the stages that require the use of soil (regularization, reinforcement and deep patches), transportation from the deposit to the application site and compaction of the material (in $m^3$) are considered.

(e)  Steel: For steel, casting, machining and welding of the material; transportation to the application site; and the execution of rigid pavement are considered (in kg).

(f)  Concrete production: Produced in a fixed plant using rock aggregates (as described above) and a binder (Portland cement), energy and fuel consumption for concrete production (in $m^3$) and subsequent transportation to the track are considered.

(g)  Asphaltic mixture production: Asphaltic mixtures are produced in fixed plants composed of rock aggregates and asphaltic binder (in tons). The steps are defined from petroleum exploration, petroleum refining, transportation to the place of use (plant), processing and subsequent transportation to the application site, finishing with the execution of the asphaltic layer.

(h)  Graduated gravel mixture production: Graduated gravel mixtures processed in a central plant are composed of rock aggregates (in $m^3$), taking into account rock blasting, aggregate crushing, transportation to the plant, aggregate mixing, transportation to the application site and the execution of the BGS layer (base layer).

(i)  Transportation: Fuel consumption is based on the average transportation distance, the type of road surface and the vehicle capacity (in ton $\times$ km). For this, petroleum exploration, petroleum refining (production of diesel in kg) and burning in use (in MJ) are considered.

(j)  Electricity: This is based on the equipment power and operating hours (in kWh).

(k)  Fuel: Petroleum exploration, petroleum refining (production of diesel in kg) and transportation between locations (production in kg and burning in MJ) is not considered.

(l) Equipment: New equipment is adopted at the beginning of the road construction, considering consumption and productivity according to DNIT guidelines.

(m) Execution: All executive stages follow the procedures used in Brazil, according to the standards, laws and executive projects. The quantification is based on the compositions of SICRO and SICRO2, covering the stages of material extraction, transportation and application.

(n) Operation: This study does not encompass the road operation stage, i.e., the use by cargo and passenger vehicles; it only encompasses only the construction stages.

(o) Maintenance: The previously described maintenance procedures are adopted (FD+RC, FD+RC+M, crack sealing and joint replacement).

(p) Restoration: Similar to the maintenance procedures, the adopted restoration alternatives are described previously: structural reinforcement, deep recycling with cement addition, Whitetopping, continuous milling with HiMA repositioning and continuous milling with repositioning (in different thicknesses depending on the need of each project).

(q) End of life: The end of the analysis period does not represent the end of the pavement structure's life span, as all have a lifespan within the analysis period (30 years). Therefore, there is no final disposal of the structure, but rather its continued use with necessary maintenance and restoration.

For the environmental study in question, the required input data for the LCI were obtained in a secondary manner from the Ecoinvent database version 3, in the quantities of the executive projects and SICRO (DNIT), and inserted into the OpenLCA 1.10.3 software for analysis and treatment. The CML 2001 method was used for the calculations, with aggregation by the weighted average and exclusion of zero values.

The environmental impact assessment was carried out through the following impact categories: global warming/climate change (GWP), acidification potential (AC), eutrophication potential (EP), freshwater aquatic ecotoxicity (FAETP), freshwater sediment ecotoxicity (FSETP), human toxicity (HTP), marine aquatic ecotoxicity (MAETP), marine sediment ecotoxicity (MSETP), photochemical oxidation (summer smog) (POCP) and terrestrial ecotoxicity (TAEP). The performance was analyzed by considering the initial intervention, the subsequent environmental performance over the 30-year life cycle of each highway and, finally, the comparison between them.

## 3. Results

### 3.1. Initial Restorations

Figure 4 shows the results obtained in the LCA for the initial restorations (with relative impact value per work) in each environmental impact category.

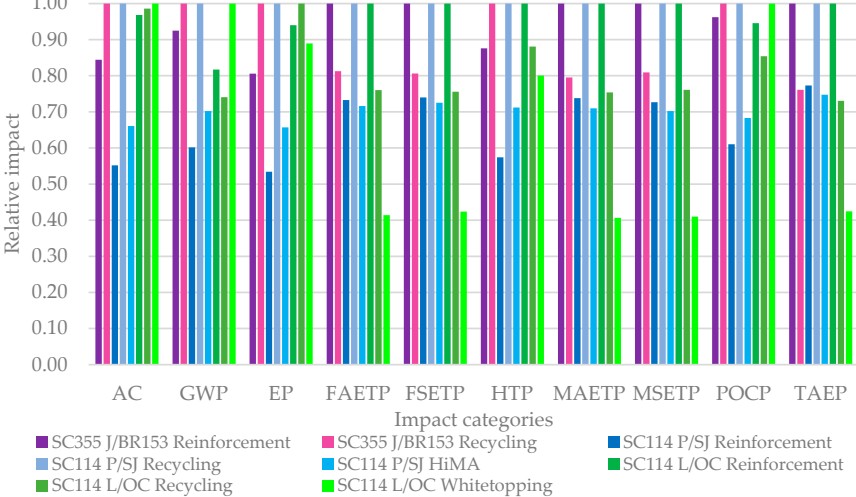

**Figure 4.** LCA initial restorations for all roads, with relative impact values.

(a)  Acidification

- SC355 J/BR153: The structural reinforcement technique shows 16% less impact than that of the technique of deep recycling with cement addition.
- SC114 P/SJ: By applying the structural reinforcement technique, there is a reduction of 45% compared to deep recycling with cement addition, and the application of a HiMA layer provides a 34% decrease in the generation of acidification impacts.
- SC114 L/OC: The Whitetopping technique shows higher generation of acidification; however, the other two proposals do not generate significant reductions in impact, with 3% for structural reinforcement and 1% for recycling. Therefore, the three techniques are initially equivalent.

One of the main contributors to acidification is the burning of fossil fuels in internal combustion engines, used in the production of aggregates, in the execution of the construction steps, in the production of asphalt binders and mainly in transportation. When comparing the recycling technique with the others, it is possible to identify greater vehicle movement for the transportation of materials and the use of different equipment, as can be observed in the Whitetopping technique; therefore, these techniques stand out in the generation of impact in the acidification category.

(b)  Climate Change

- C355 J/BR153: The application of structural reinforcement, compared to deep recycling with cement addition, reduces the impact related to climate change by 8%.
- SC114 P/SJ: The recycling technique shows the highest impact among the analyzed techniques, with the structural reinforcement and HiMA allowing for a reduction of 40% and 30%, respectively.
- SC114 L/OC: Unlike acidification, for the category of climate change, applying the structural reinforcement and recycling instead of Whitetopping results in a significant reduction in impact, being 18% and 26%, respectively.

The amount of used material; transportation; the execution of the layers, which consume a high amount of diesel; and the production of the asphalt binder significantly contribute to the generation of this impact.

(c)  Eutrophication

The production of aggregate, especially the crushing and screening phase, followed by the use of fuel for equipment and transportation, results in a higher eutrophication impact. This was identified in studies in which granular layers had the greatest impact on the eutrophication category [46], explaining the results found in this study, in which the recycling technique has the highest impact contributions.

- SC355 J/BR153: With the application of structural reinforcement, there is a 19% reduction in impact.
- SC114 P/SJ: By interfering in a smaller number in granular layers, the structural reinforcement and HiMA present reductions of 47% and 34%, respectively, compared to deep recycling.
- SC114 L/OC: Despite the structural reinforcement presenting a thick thickness (15 cm), there is a reduction of 6%, and for Whitetopping, there is a reduction of 11% compared to deep recycling.

(d)  Ecotoxicity

- SC355 J/BR153: Recycling presents a 20% reduction in impact compared to structural reinforcements.
- SC114 P/SJ: Reinforcements and HiMA generate 26% and 28% less environmental impact, respectively, in the ecotoxicity category compared to deep recycling.
- SC114 L/OC: The case study with the thickest coating in the structural reinforcement technique, which provides the largest ecotoxicity impact, by incorporating

recycling and Whitetopping, enables a reduction of 24% and 59% compared to reinforcements.

It becomes clear that there is significant interference of asphalt binder production, which is used in asphalt coating layers, bonding paint, priming, surface repairs and deep repairs. A higher proportion of use is identified in the structural reinforcement techniques; however, in SC114 P/SJ, the thickness of the reinforcement layer is lower than that of the coating layer after recycling, explaining why it presents a higher impact.

(e)    Human toxicity

In road work, it has been identified that the use of diesel as a fuel for the operation of machinery and the production of asphalt binder are systems that most influence the quantification of this environmental impact.

- SC355 J/BR153: The application of reinforcements reduces impact by 12%.
- SC114 P/SJ: Due to the application of a thinner coating than those of other works, reinforcements and HiMA allow reductions of 43% and 29%, respectively, compared to deep recycling.
- SC114 L/OC: Deep recycling and Whitetopping show reductions of 12% and 20%, respectively, compared to structural reinforcement.

(f)    Photochemical oxidation

- SC355 J/BR153: The two techniques have a very similar impact, with reinforcements being only 4% lower.
- SC114 P/SJ: The use of reinforcements reduces photochemical oxidation impact by 39% and HiMA by 32% compared to recycling.
- SC114 L/OC: Reinforcements and Whitetopping generate a similar amount of impact, whereas applying recycling has a reduction of 15%.

The production of asphalt binders, the consumption of diesel by the machinery, the production of aggregates, and transportation contribute a significant portion to the generation of impacts from the category of photochemical oxidants, being responsible for respiratory problems in the population that is living in the vicinity of their generation [8].

It is evident that the production of bituminous binders and asphalt mixtures has a significant impact on the construction of roads, making it essential to use techniques that reduce emissions into the environment and to adopt alternative or recycled materials. However, evaluating the best alternative only via the initial restoration technique is not recommended, as each one has different behaviors regarding durability that are dependent on traffic and road management policy; therefore, it is essential to study a certain life cycle period, which will be presented in a sequence per work.

*3.2. Road SC355 J/BR153*

3.2.1. Structural Reinforcement

In search of the evaluation of the entire studied life cycle, Figure 5 presents the contributions of each intervention applied throughout the LCA period for the initial restoration technique, "Structural Reinforcement".

The categories of impact acidification, climate change and eutrophication show the same percentages of contribution in each stage, with the initial reinforcement with 10 cm, with FC+RC90 being the one that contributes the most with 23% each, followed by FD+RC+M with a 3% impact in relation to reinforcement and FD+RC with 1%. The contribution of FD+RC+M and FD+RC results in the same contributions in all the analyzed impact categories, precisely because they present the same technique and percentage throughout all applications.

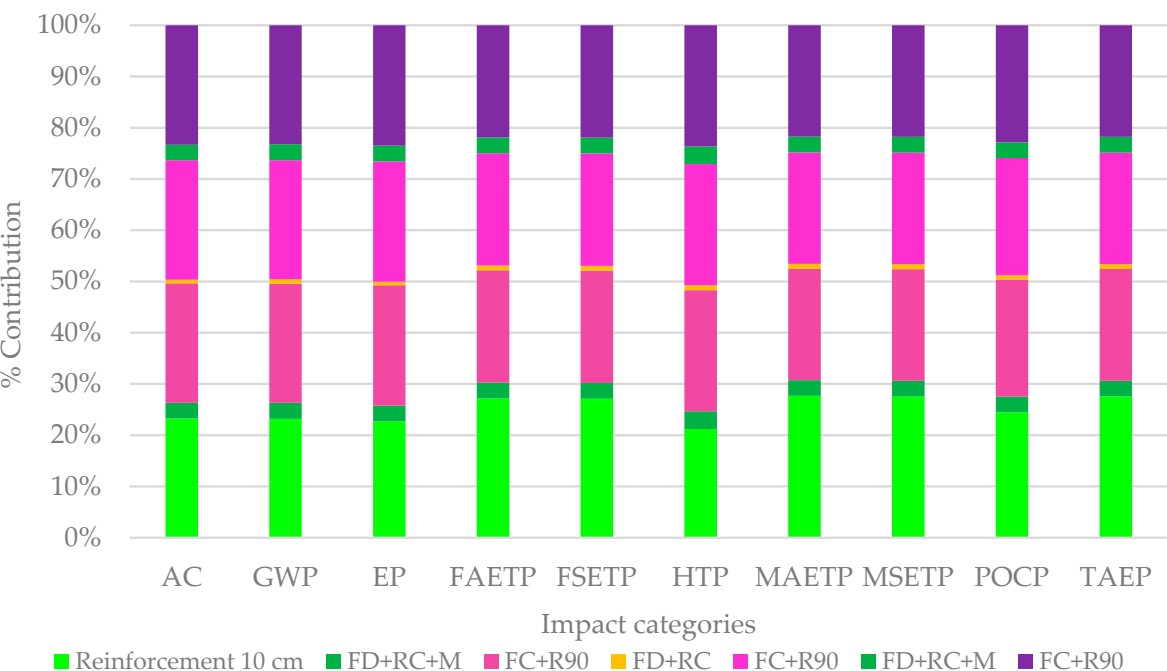

**Figure 5.** Contribution of each intervention applied over the 30 years of LCA to the initial restoration technique, "Structural reinforcement", for SC355 J/BR153.

For the ecotoxicity category, the influence of the thickness of the asphalt coating on the impact contribution is evident, in which the structural reinforcement with 10 cm is equivalent to 28% of the impact. In addition, FC+RC90 has 22% of the impact, despite having only 1 cm of difference, so the amount of material produced in one kilometer of length becomes significant. In relation to human toxicity, the greatest contribution is identified when executing FC+RC90 with 24%, followed by structural reinforcement (10 cm) with 21%. The increase in contribution for the milling stage is attributed to the consumption of diesel by equipment and vehicles for the transportation of waste and raw materials, which stands out compared to structural reinforcement.

For photo-chemical oxidation, where binder production is the largest contributor, it is noted that reinforcements and FC+RC90 contribute 25% and 23%, respectively, which are close values considering the thickness of the layers with a difference of 1 cm. The initial intervention, the restoration type, represents between 21% and 28% of the impact, depending on the evaluated category. Impacts on the ecosystem, such as ecotoxicity, are the largest due to the thickness of the applied asphalt layer, and human toxicity is the lowest in the early years, as milling generates more impact than that of the structural reinforcement itself. Marine and aquatic sediment ecotoxicity results in an impact in the order of $8 \times 10^4$ kg 1,4-DCB$_{Eq}$, whereas freshwater aquatic and sediment ecotoxicity generates around $1 \times 10^3$ kg 1,4-DCB$_{Eq}$.

### 3.2.2. Deep Recycling

Figure 6 details the contributions of each intervention applied to SC355 J/BR153 throughout the evaluation period for the initial restoration, "Deep Recycling".

The acidification and eutrophication impact categories present the same percentages of contribution at each stage, with recycling contributing 57% and FC+RC75 contributing 18% each, followed by FD+RC+M with a 6% impact in relation to reinforcement and FD+RC with 1%. Due to the amount of needed aggregate and diesel consumption in the execution and transportation, the contributions of FD+RC+M and FD+RC result in the same contributions in all analyzed impact categories, precisely because they present the same technique and percentage throughout all applications.

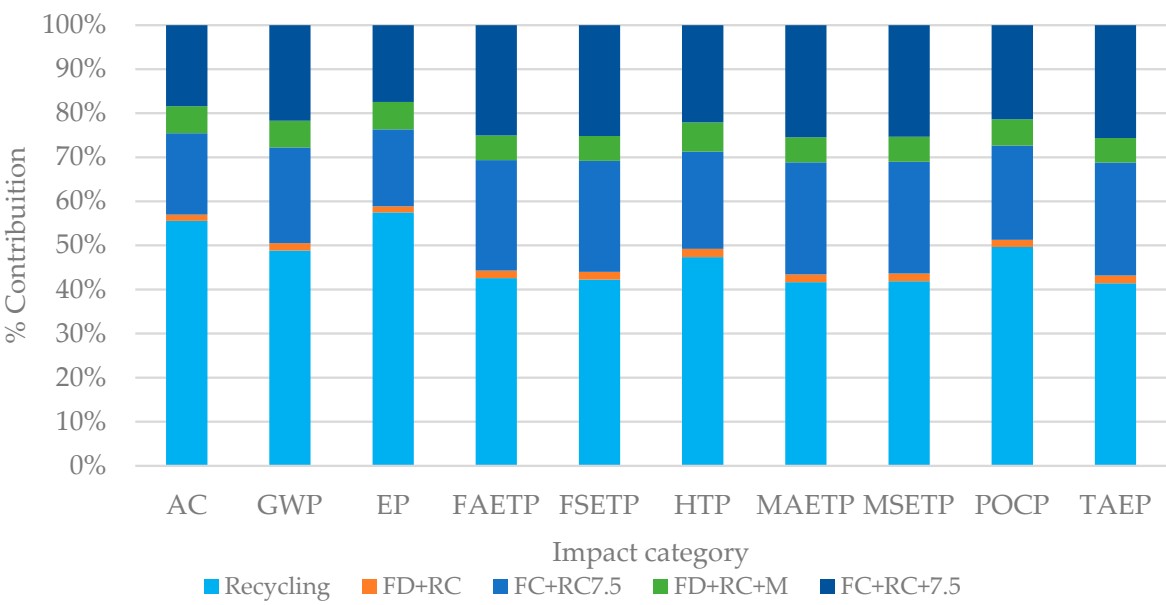

**Figure 6.** Contribution of each intervention applied over the 30 years of LCA to the initial restoration technique, "Deep recycling", for SC355 J/BR153.

In relation to the climate change category, recycling contributes 49%, and FC+RC75 contributes 22%. For the ecotoxicity category, the influence of the asphalt coating thickness on the contribution of impact is evident, in addition to the use of binder in the surface treatment in the stage of deep recycling with the addition of cement, which accounts for 42% of impact, and FC+RC75 accounts for 25%. For human toxicity and photo chemical oxidation, the largest contribution is identified when deep recycling is executed with 48%, followed by FC+RC75 with 22%. The largest consumption in the execution of recycling is attributed to the consumption of diesel by equipment and vehicles for the transportation of raw materials and the production of asphalt binder, which stands out compared to continuous milling.

The initial intervention, the restoration type, represents between 41% and 57% of the impact, depending on the evaluated category. Recycling has a greater influence on acidification, eutrophication and photo-chemical oxidation, with more than 50% of the contribution throughout the analysis period, so it is very important to plan the movement of materials and the execution, due to fuel consumption and the production of aggregates and binders. The used transportation distance is one of the main factors in the high incidence of impacts.

Comparing the impacts generated between the initial structural reinforcement approach and deep recycling, it is evident that, at first, in the first intervention type, restoration, deep recycling has a greater impact on categories that are not managed by the production of asphalt binder. Structural reinforcement becomes more harmful when the impact categories are dependent on the production of asphalt binder. However, when planning interventions over 30 years, the behavior of environmental impact generation changes. The recycling approach becomes approximately 50% less impactful than the structural reinforcement, which is due to the number of interventions required over the period, the behavior of the adopted layers and the thickness of the coating that is used for the reconstitutions. Thus, the reinforcement approach has higher consumption of aggregates, asphalt binders, diesel for transportation and execution of layers. For SC355 J/BR153, it is evident that the "deep recycling" approach is environmentally more satisfactory in relation to the "structural reinforcement" approach when considering the 30-year study period.

### 3.3. Road SC114 P/SJ

#### 3.3.1. Structural Reinforcement

Figure 7 displays the contributions of each intervention applied to SC114 P/SJ throughout the analysis period, considering the initial restoration, "Structural Reinforcement". The contribution ratio between the impact categories are very similar. On average, reinforcement contributes approximately 17%, FC+RC90 generates 25% of the impact in each category, and FD+RC+M and FD+RC have low interference, with 3% and 1%, respectively.

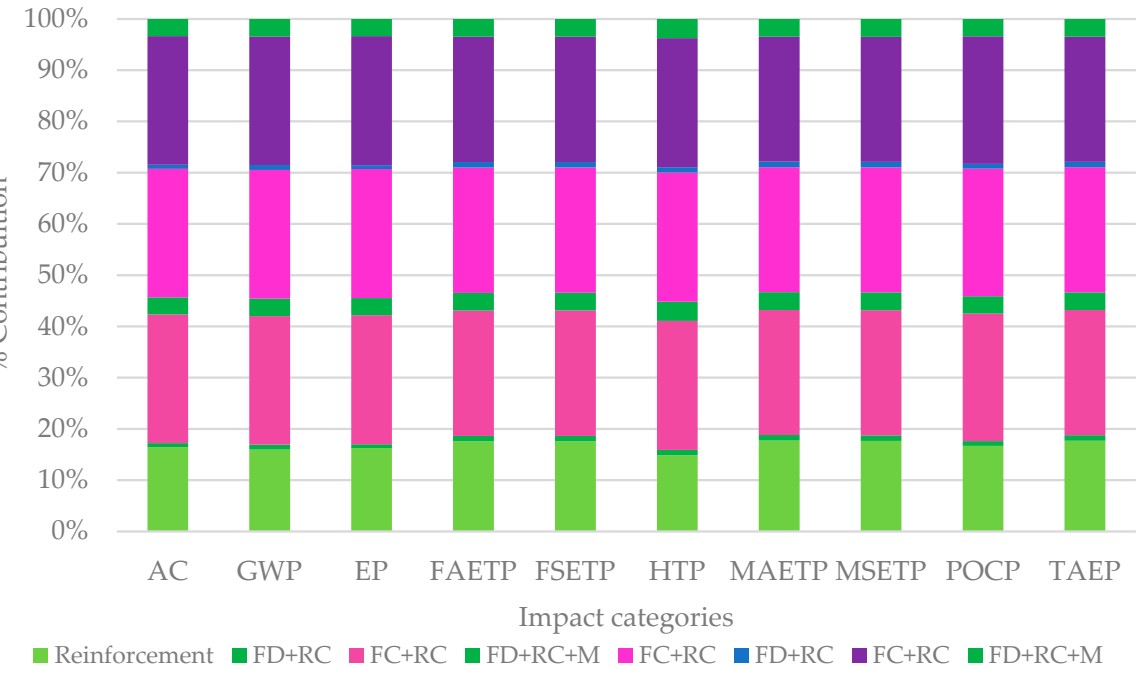

**Figure 7.** Contribution of each intervention applied over the 30 years of LCA to the initial restoration technique, "Structural reinforcement", for SC114 P/SJ.

The largest contribution in FC+RC90 comes from the amount of asphalt mixture applied in this technique, in which a 9 cm thickness is used, whereas the initial technique, "reinforcement", is executed with 5 cm. The production of aggregate and the production of asphalt binder, combined with fuel consumption in executing the layers and transporting the materials, increase the contribution of generated impacts.

#### 3.3.2. Deep Recycling

Figure 8 details the contributions of the interventions considering deep recycling with cement addition. The deep recycling stage generates a 57% impact in the acidification category. FC+RC70 contributes 18%, FD+RC+M contributes 6%, and FD+RC contributes only 2% of the total throughout the analysis period, making it clear that the fuel consumption that is used by the equipment for the recycling execution, along with the production of aggregates and binders to compose the layers and transport, significantly impact the execution of the road.

Regarding ecotoxicity, the largest interference occurs in the asphalt layers. Recycling contributes 43%, and FC+RC90 contributes 25% each. For the climate change category, recycling contributes 50%, and FC+RC90 contributes 21%. Due to the amount of aggregate required and the diesel consumption in execution and transport, recycling contributes 59% of the impact in eutrophication.

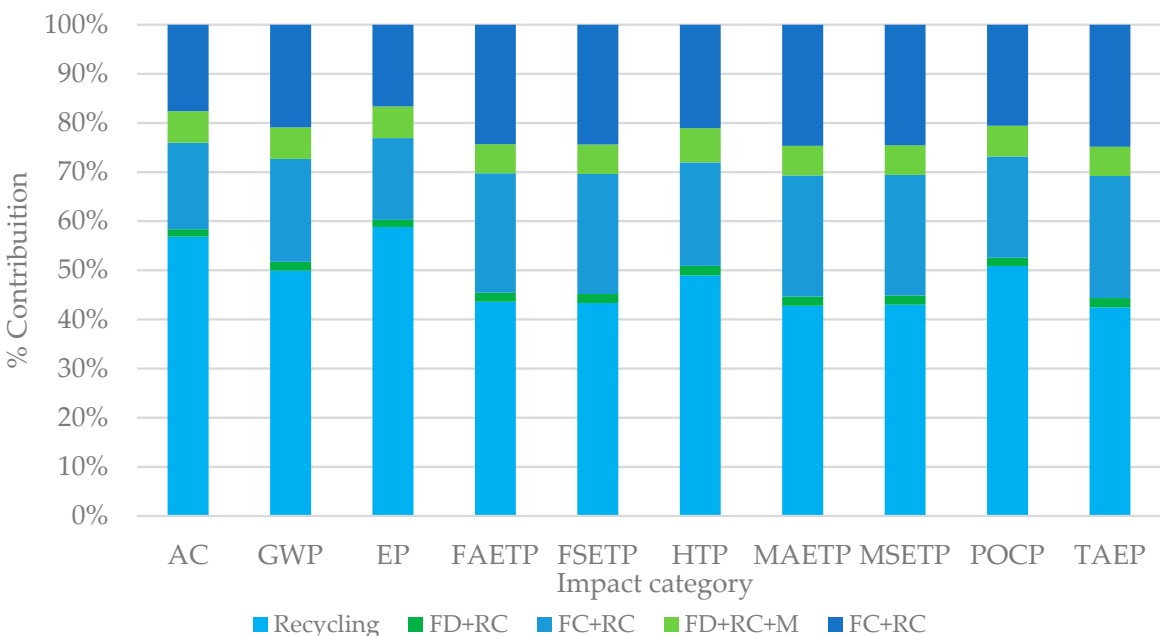

**Figure 8.** Contribution of each intervention applied over the 30 years of LCA to the initial restoration technique, "Deep recycling", for SC114 P/SJ.

Regarding human toxicity and photochemical oxidation, the highest contribution is identified when deep recycling is performed with 50%, followed by FC+RC90 with 21%. The largest consumption in the execution of recycling is attributed to the consumption of diesel by the equipment and vehicles for the transportation of raw materials and the production of asphalt binder, which stands out in comparison to continuous milling.

The initial intervention, the restoration type, represents between 42% and 59% of the impact for the evaluated categories. Recycling has a greater impact on eutrophication, acidification, photochemical oxidation and climate change, with over 50% of the contribution throughout the analysis period. Given this, it is very important to plan the movement of materials and the execution, due to the consumption of fuels and the production of aggregates and binders.

### 3.3.3. HiMA

Figure 9 shows the contributions of the HiMA approach to SC114 P/SJ. Guaranteeing that, throughout the study period, millings with recompositions are executed with the same material as that of the initial intervention, it was identified that this stage contributes to approximately 24% of the impact generated with each application. This shows the importance of dimensioning the thickness and maintaining the road so that intermediate interventions of greater size are not necessary.

FD+RC and FD+RC+M contribute 1% and 5%, respectively, which are values that are considered small compared to the contributions of the other interventions, but they cannot be ignored. Analyzing ecotoxicity, it was determined that the greatest interference is the asphalt layers. FC+RCH contributes 23% each, and structural reinforcement (5 cm thickness) contributes 24%. The initial intervention, the restoration type, represents only 24% of the impact for the evaluated categories.

Comparing the impacts generated by the initial structural reinforcement approach, deep recycling and HiMA, it becomes evident that, at first, in the first restoration intervention, deep recycling has a greater impact in all categories. However, when planning the interventions over 30 years, the behavior of the environmental impact generation changes. The recycling approach becomes approximately 45% less impactful than structural reinforcement and 30% less impactful than HiMA, which is due to the number of interventions that are necessary over the period (reinforcement with eight interventions, HiMA with

seven interventions and recycling with five interventions), the behavior of the adopted layers and the thickness of the coating that is used for the reconstructions. With this, the reinforcement approach has a higher consumption of aggregates, asphalt binders, diesel for transportation and execution of the layers. The HiMA approach becomes 20% less environmentally impactful than reinforcement. For SC114 P/SJ, it becomes evident that the "deep recycling" approach is more environmentally satisfactory in relation to the "structural reinforcement" and "HiMA" approaches when considering the 30-year study period.

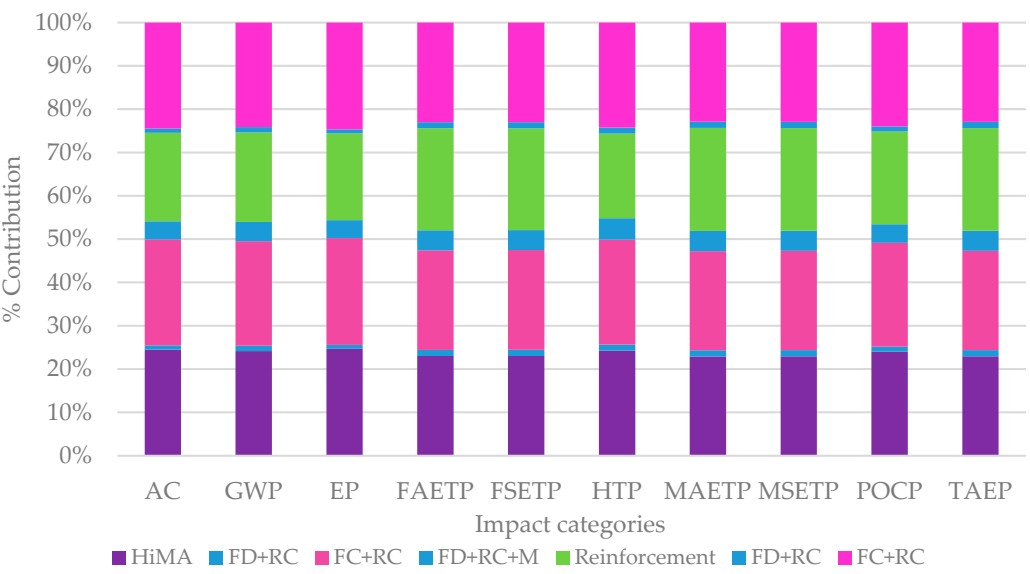

**Figure 9.** Contribution of each intervention applied over the 30 years of LCA for the "HiMA" initial restoration technique for SC114 P/SJ.

### 3.4. Road SC114 L/OC

### 3.4.1. Structural Reinforcement

SC114 L/OC is the highway that presents the highest flow of freight vehicles and the highest daily average of vehicles; therefore, interventions occur in shorter periods of time than they do in other highways. Figure 10 shows the contributions to initial restoration and structural reinforcement.

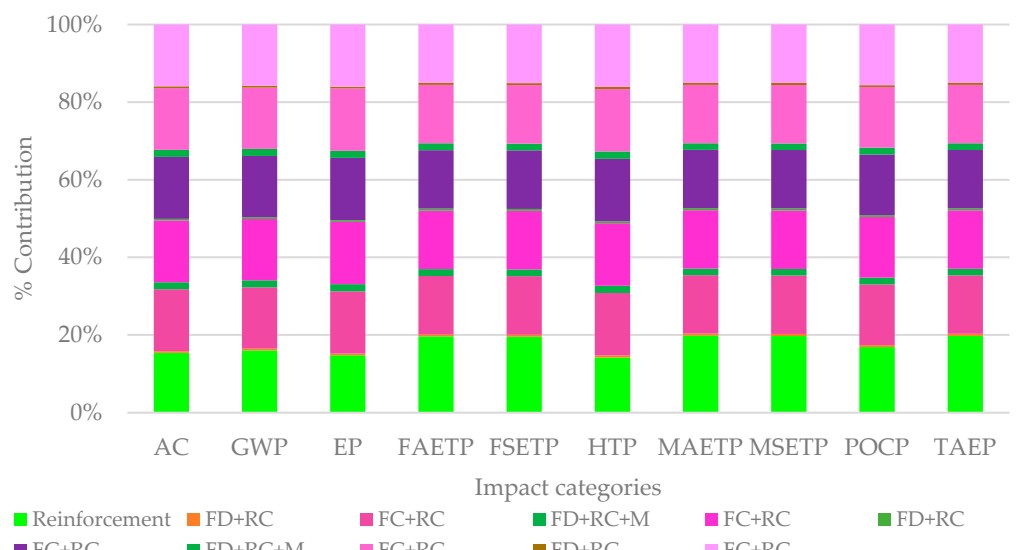

**Figure 10.** Contribution of each intervention applied over the 30 years of LCA to the initial restoration technique, "Structural reinforcement", for SC114 L/OC.

The contribution proportion between impact categories is very similar, on average. Reinforcement contributes between 14% and 20%, FC+RC140 generates 15.5% of the impact in each category, and FD+RC+M and FD+RC present low interference, with 1.8% and 0.5%, respectively. The balance among the restorations is mainly due to the thickness of the coating (15 cm with reinforcement and 14 cm with continuous milling). Regarding the proximity between the impact categories, it is justified by the balance between the amount of aggregate that is necessary for the execution of the layers and the coating structure, as well as fuel consumption for the operation of the equipment and the transportation of materials. This highway presents a subgrade reinforcement layer, which is not used in the previous ones.

Due to its higher freight flow compared to the other studied highways, it requires a thicker structure and more interventions in a shorter period of time in order to support efforts and to ensure rolling quality and user safety. Therefore, the generation of environmental impacts is well distributed over the 30 years, requiring 11 interventions of restoration and maintenance, which may not be an adequate technique for local traffic.

3.4.2. Deep Recycling

Figure 11 presents the contributions related to the interventions for the initial restoration, deep recycling.

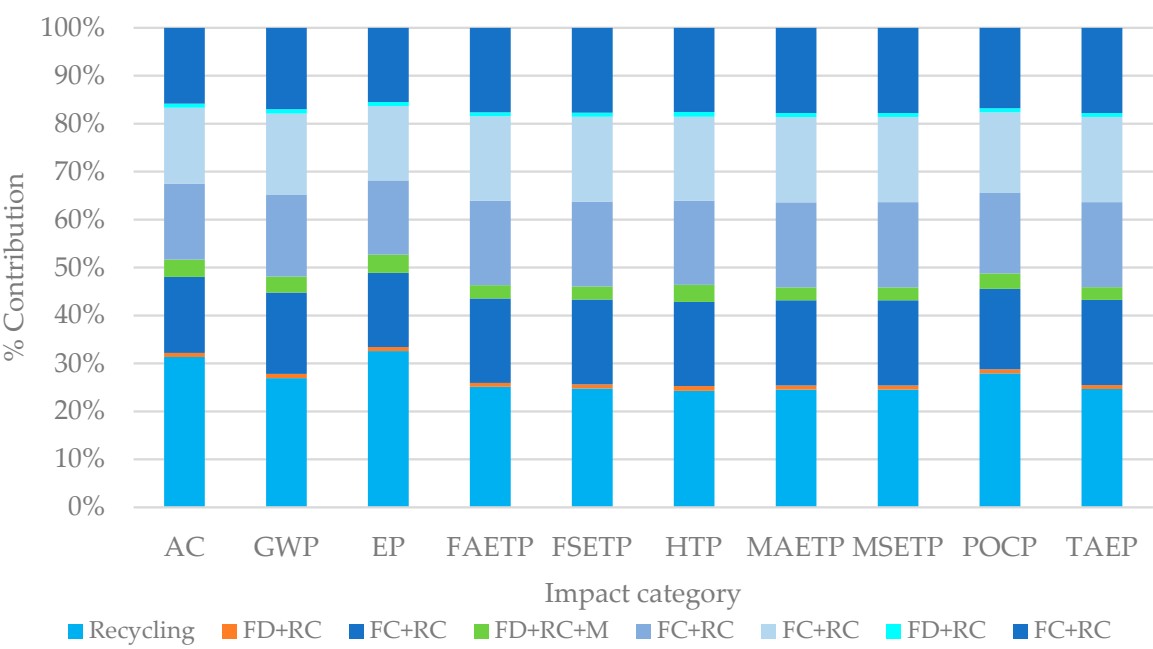

**Figure 11.** Contribution of each intervention applied over the 30 years of LCA to the initial restoration technique, "Deep recycling", for SC114 L/OC.

The deep recycling stage has an impact of 32% in the acidification category. FC+RC110 contributes 16%, FD+RC+M contributes 4%, and FD+RC contributes only 1% of the total throughout the analysis period. The fuel consumption used by the equipment for the recycling, along with the production of aggregates and binders to comprise the layers and the transportation, significantly impacts the execution of the road. However, attention must be paid to the coating thickness that is used in continuous milling restorations, which mainly contribute to the amount of binder and aggregates required for construction.

Regarding ecotoxicity, the greatest interference comes from the asphalt layers. However, recycling contributes 25%, and FC+RC110 contributes 18% each. For the climate change category, recycling contributes 27%, and FC+RC110 contributes 17%. Due to the amount of aggregate required and the diesel consumption in execution and transportation, recycling contributes 33% of the impact on eutrophication. For human toxicity, the greatest

contribution is identified for deep recycling with 24%, followed by FC+RC110 with 18%. The highest consumption in deep recycling execution is attributed to diesel consumption by equipment and vehicles for the transportation of raw materials and the production of asphalt binder, which stands out compared to continuous milling. It is identified that, in photochemical oxidation, there is the same behavior, with recycling contributing 28%.

As mentioned earlier, for SC114 L/OC, with high traffic under consideration, there is a distribution over the years of necessary interventions. The initial intervention, recycling, represents, on average, 25% of the impacts for the evaluated categories. Considering that, over the 30 years, eight interventions are necessary, it can be said that it is a considerable percentage for the start of the work. On the other hand, throughout the evaluation period, the necessary interventions generate lower impacts and present a lower number of interventions compared to reinforcement, which reduces discomfort for users and residents in the surroundings.

### 3.4.3. Structural Reinforcement

The Whitetopping restoration technique has a high environmental impact in its application, as it requires the execution of a 22 cm Portland cement concrete layer, in addition to deep repairs and the regularization of the shoulder and wheel tracks with the application of an asphalt mixture. Based on the simulations, throughout the 30 years, Whitetopping generates more than 90% of the total impact (Figure 12). However, attention must be paid to the boundaries of the study; depending on what is considered, these impacts may change.

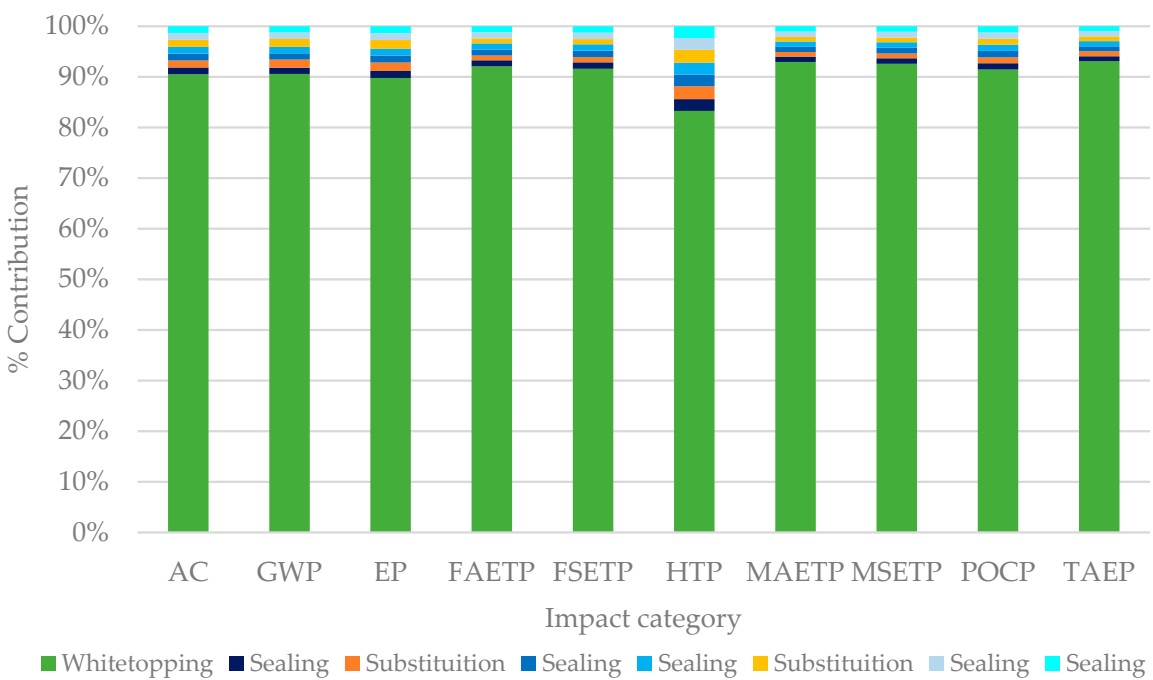

**Figure 12.** Contribution of each intervention applied over the 30 years of LCA to the "Whitetopping" initial restoration technique for SC114 L/OC.

Only the human toxicity impact category shows a contribution of 83% for the Whitetopping technique, with an increase in crack sealing contribution from 1.3% to 2.3% and and increase in joint replacement from 1.3% to 2.5%, showing the interference of diesel consumption and binders for the execution of these interventions.

Comparing the impacts generated by the initial structural reinforcement approach, deep recycling, and Whitetopping, it is evident that, initially, in the first restoration intervention, deep recycling has a greater impact when analyzed in the eutrophication category. When checking the categories of acidification, climate change and photochemical oxidation, the Whitetopping technique stands out. However, when evaluating ecotoxicity and human toxicity, reinforcement is the technique that initially has the greatest impact. However,

when planning interventions over 30 years, the environmental impact generation behavior changes. Reinforcement becomes the approach with the greatest impact in all categories. Recycling presents a significant reduction of around 40%, and Whitetopping presents a reduction of more than 80%. However, each category presents different variations.

For the categories of ecotoxicity, recycling shows a reduction of around 40%, and Whitetopping shows a reduction of 90% compared to reinforcement, which is justified by the lesser application of asphalt coating in the restoration and maintenance techniques used over time. For the other categories, the reduction is 50% when recycling is applied, and it is 80% with Whitetopping, proving the performance of semi-rigid and rigid structures both structurally and environmentally. For SC114 L/OC, it is evident that the "Whitetopping" approach is environmentally more satisfactory compared to the "structural reinforcement" approach, with "Deep Recycling" being a second option when considering the 30-year study period.

## 4. Discussion

Given the climate change that we constantly live in, it is necessary to think and act to minimize the impact that human actions have on the environment. To do this, all engineering works must have studies to identify the environmental impacts that they generate.

### 4.1. Initial Restorations

The developed study results in the following initial impacts (Table 5).

**Table 5.** Impacts generated by initial restorations.

| CI | AC | GWP | EP | FAETP | FSETP | HTP | MAETP | MSETP | POCP | TAEP |
|---|---|---|---|---|---|---|---|---|---|---|
| SC355 JBR153 | | | | | | | | | | |
| RE | $9.06 \times 10^2$ | $8.32 \times 10^4$ | $1.44 \times 10^2$ | $2.00 \times 10^2$ | $4.34 \times 10^2$ | $2.84 \times 10^4$ | $2.12 \times 10^4$ | $2.31 \times 10^4$ | $1.40 \times 10^1$ | $2.79 \times 10^1$ |
| RC | $1.07 \times 10^3$ | $9.00 \times 10^4$ | $1.78 \times 10^2$ | $1.62 \times 10^2$ | $3.50 \times 10^2$ | $3.24 \times 10^4$ | $1.68 \times 10^4$ | $1.87 \times 10^4$ | $1.45 \times 10^1$ | $2.12 \times 10^1$ |
| SC114 PSJ | | | | | | | | | | |
| RE | $6.10 \times 10^2$ | $5.40 \times 10^4$ | $9.84 \times 10^1$ | $1.20 \times 10^2$ | $2.64 \times 10^2$ | $1.95 \times 10^4$ | $1.27 \times 10^4$ | $1.39 \times 10^4$ | $9.02$ | $1.67 \times 10^1$ |
| RC | $1.11 \times 10^3$ | $8.98 \times 10^4$ | $1.84 \times 10^2$ | $1.64 \times 10^2$ | $3.57 \times 10^2$ | $3.39 \times 10^4$ | $1.72 \times 10^4$ | $1.92 \times 10^4$ | $1.48 \times 10^1$ | $2.16 \times 10^1$ |
| HM | $7.31 \times 10^2$ | $6.30 \times 10^4$ | $1.21 \times 10^2$ | $1.18 \times 10^2$ | $2.59 \times 10^2$ | $2.42 \times 10^4$ | $1.22 \times 10^4$ | $1.35 \times 10^4$ | $1.01 \times 10^1$ | $1.62 \times 10^1$ |
| SC114 LOC | | | | | | | | | | |
| RE | $1.10 \times 10^3$ | $1.14 \times 10^5$ | $1.70 \times 10^2$ | $2.79 \times 10^2$ | $6.07 \times 10^2$ | $3.74 \times 10^4$ | $2.98 \times 10^4$ | $3.25 \times 10^4$ | $1.85 \times 10^1$ | $3.92 \times 10^1$ |
| RC | $1.12 \times 10^3$ | $1.03 \times 10^5$ | $1.81 \times 10^2$ | $2.12 \times 10^2$ | $4.58 \times 10^2$ | $3.29 \times 10^4$ | $2.24 \times 10^4$ | $2.48 \times 10^4$ | $1.67 \times 10^1$ | $2.87 \times 10^1$ |
| WH | $1.14 \times 10^3$ | $1.40 \times 10^5$ | $1.61 \times 10^2$ | $1.16 \times 10^2$ | $2.57 \times 10^2$ | $2.99 \times 10^4$ | $1.21 \times 10^4$ | $1.33 \times 10^4$ | $1.96 \times (-10^1)$ | $1.67 \times 10^1$ |

RE (Reinforcement), RC (Recycling), HM (HiMA), WH (Whitetopping); Higher impact – Lower impact .

When comparing the recycling technique with others, a higher movement of vehicles is observed for the transportation of materials and the use of different equipment, as can also be observed in the Whitetopping approach. Therefore these techniques stand out in generating impacts in the acidification category due to the burning of fossil fuels in combustion engines used in the production of aggregates, in the execution of the work steps, in the production of asphalt binders and mainly in transportation. For SC355 J/BR153, recycling presents 18% more acidification than reinforcement, whereas for SC114 P/SJ, this difference is 82% (due to the low thickness of the reinforcement of only 5 cm, and the recycling coating is 8 cm). The values found for the three approaches for SC114 L/OC are equivalent.

The same behavior is observed in the climate change categories; however SC114 L/OC requires a larger amount of aggregate and mainly Portland cement, as well as asphalt mixture for filling the shoulders and wheel tracks, resulting in a greater impact, with the Whitetopping technique being 23% more impactful than reinforcement and 36% more impactful than recycling. Compared with other works, the reinforcement of SC114 L/OC (reinforcement of 15 cm) presents 37% more impact than that of SC355 J/BR153 (with reinforcement of 10 cm) and 111% more impact compared to that of SC114 P/SJ (with reinforcement of 5 cm), thus showing the high impact that the thickness of the asphalt layer has on the impact category.

The production of aggregate, mainly the crushing and sorting phase, followed by the use of fuel for equipment and transportation, results in a higher eutrophication impact. This was identified in studies in which the granular layers are the ones that had the greatest impact on the eutrophication category [46], justifying the findings in this study, in which the recycling technique presents the highest contribution of impacts in the three analyzed works. Recycling generates an impact of eutrophication in the order of $1.8 \times 10^2$ kg $PO_{4Eq}$ for the three works, and structural reinforcement varies depending on the thickness of the coating layer, with $9.8 \times 10^2$ kg $PO_{4Eq}$ (thickness of 5 cm), $1.4 \times 10^2$ kg $PO_{4Eq}$ (thickness of 10 cm) and $1.8 \times 10^2$ kg $PO_{4Eq}$ (thickness of 15 cm).

Higher impact generation in the ecotoxicity category is identified when the structural reinforcement technique is applied, being approximately 30% higher; however, in SC114 P/SJ, the thickness of the reinforcement layer is lower than that of the recycling coating layer, justifying the presentation of greater impacts (35% higher than that of reinforcement). The production of the asphalt binder, which is used in the asphalt coating layers, bonding paint, priming, surface treatment, superficial repairs and deep repairs, is the main contributor to this category.

In road construction, the use of diesel as a fuel for machinery and the production of bitumen are the systems that most influence the quantification of the environmental impact on human toxicity. In SC355 J/BR153 and SC114 P/SJ, the execution of the recycling layer is the biggest contributor (14% and 74%, respectively, higher than reinforcement), but when working with high-thickness structural reinforcement (15 cm in SC114 L/OC), the production of asphalt binder is more impactful than the execution itself (14% higher compared to recycling).

The production of asphalt binders, together with the consumption of diesel by machinery, the production of aggregates, and transportation, contribute significantly to the generation of impacts in the category of photochemical oxidants. In the evaluated works, SC355 J/BR153 and SC114 L/OC, which have very similar thicknesses of reinforcement and recycling coatings, result in similar impacts (4% and 11%). However, in SC114 P/SJ, in which there is a significant difference in thickness, the recycling technique results in a 64% impact.

### 4.2. Interventions over the 30 Years of Analysis

When evaluating the best alternative based only on the initial restoration technique, it is not recommended because each one presents different behaviors in terms of durability that depend on traffic and road management policies. Therefore, it is essential to study over a certain life cycle period. Table 6 presents the accumulated impacts over 30 years of analysis.

However, by assessing the total impacts generated by each approach over the 30-year analysis period, it was found that reinforcement presents the highest impact. For SC355 J/BR153, recycling results in approximately 48% less impact than reinforcement. For SC114 P/SJ, recycling is able to reduce the total impact by 45%, and HiMA is able to reduce it by 25%. Finally, for SC114 L/OC, recycling enables a 40% emission, and Whitetopping has 90% less impact than reinforcement.

It is evident that the use of rigid structures such as Whitetopping or semi-rigid structures such as deep recycling with the addition of Portland cement generates less environmental impact when compared to the use of flexible structures that consume a large amount of asphalt binder and require interventions at shorter time intervals. It should be noted that the operation stages, interference in vehicle flow and user expenses, among other factors that can contribute to emissions, are not considered.

**Table 6.** Impacts generated over the 30-year analysis cycle.

| CI | AC | GWP | EP | FAETP | FSETP | HTP | MAETP | MSETP | POCP | TAEP |
|---|---|---|---|---|---|---|---|---|---|---|
| | | | | | SC355 JBR153 | | | | | |
| RE | $3.89 \times 10^3$ | $3.59 \times 10^5$ | $6.32 \times 10^2$ | $7.34 \times 10^2$ | $1.60 \times 10^3$ | $1.34 \times 10^5$ | $7.65 \times 10^4$ | $8.40 \times 10^4$ | $5.72 \times 10^1$ | $1.01 \times 10^2$ |
| RC | $1.93 \times 10^3$ | $1.84 \times 10^5$ | $3.10 \times 10^2$ | $3.81 \times 10^2$ | $8.27 \times 10^2$ | $6.84 \times 10^4$ | $4.04 \times 10^4$ | $4.47 \times 10^4$ | $2.93 \times 10^1$ | $5.12 \times 10^1$ |
| | | | | | SC114 PSJ | | | | | |
| RE | $3.72 \times 10^3$ | $3.38 \times 10^5$ | $6.05 \times 10^2$ | $6.86 \times 10^2$ | $1.50 \times 10^3$ | $1.31 \times 10^5$ | $7.16 \times 10^4$ | $7.88 \times 10^4$ | $5.41 \times 10^1$ | $9.45 \times 10^1$ |
| RC | $1.95 \times 10^3$ | $1.80 \times 10^5$ | $3.13 \times 10^2$ | $3.77 \times 10^2$ | $8.23 \times 10^2$ | $6.93 \times 10^4$ | $4.02 \times 10^4$ | $4.46 \times 10^4$ | $2.91 \times 10^1$ | $5.09 \times 10^1$ |
| HM | $2.99 \times 10^3$ | $2.61 \times 10^5$ | $4.91 \times 10^2$ | $5.12 \times 10^2$ | $1.12 \times 10^3$ | $9.96 \times 10^4$ | $5.33 \times 10^4$ | $5.87 \times 10^4$ | $4.21 \times 10^1$ | $7.05 \times 10^1$ |
| | | | | | SC114 LOC | | | | | |
| RE | $7.14 \times 10^3$ | $7.12 \times 10^5$ | $1.15 \times 10^3$ | $1.42 \times 10^3$ | $3.10 \times 10^3$ | $2.63 \times 10^5$ | $1.50 \times 10^5$ | $1.65 \times 10^5$ | $1.09 \times 10^2$ | $1.98 \times 10^2$ |
| RC | $3.57 \times 10^3$ | $3.84 \times 10^5$ | $5.55 \times 10^2$ | $8.45 \times 10^2$ | $1.85 \times 10^3$ | $1.36 \times 10^5$ | $9.14 \times 10^4$ | $1.01 \times 10^5$ | $5.98 \times 10^1$ | $1.16 \times 10^2$ |
| WH | $1.26 \times 10^3$ | $1.54 \times 10^5$ | $1.79 \times 10^2$ | $1.26 \times 10^2$ | $2.80 \times 10^2$ | $3.59 \times 10^4$ | $1.30 \times 10^4$ | $1.44 \times 10^4$ | $2.14 \times 10^1$ | $1.79 \times 10^1$ |

RE (Reinforcement), RC (Recycling), HM (HiMA), WH (Whitetopping); Higher impact – Lower impact .

Concrete is an economical paving solution and consumes a minimum amount of materials, energy and resources for construction, maintenance and rehabilitation throughout its lifetime, increasing its sustainability and reducing fuel consumption by vehicles [58]. Energy consumption is higher when applying asphalt mixtures, and the impact of global warming is higher when using Portland cement concrete; however, many factors can affect the results of LCA, such as system limits, the quality and source of inventory data, inconsistent pavement projects and geographical locations [40,45].

The use of Portland cement concrete pavements reduces $CO_{2eq}$ emissions by 9.2%, whereas the use of asphalt mixtures causes a 19.1% increase in these emissions [16]. Thus, it is evident that, not only in construction, but also during use, concrete pavement is environmentally advantageous.

Acidification is the increase in the acidity level of soil, water and air, the main contributors of which are anthropogenic activities, such as the combustion of fossil fuels in industrial activities and vehicles. With the large amount of aggregate used in restorations with structural reinforcement, there is significant fuel consumption both for extracting the aggregates, for transportation between the extraction site and the application and for executing the layers. Another process with a high impact is the production of asphalt binders, making it evident that the coating layer contributes to the generation of impacts, and therefore, the structural reinforcement approach is the most impactful.

The climate change impact category represents the increase in the average temperature of the planet due to anthropogenic emissions of greenhouse gases, with the main sources being the combustion of fossil fuels such as oil, coal and natural gas, receiving greater contributions from the reinforcement approach over the 30-year analysis. This is justified by the execution stages of the pavement layers, for which equipment consumes a large amount of diesel for its operation; by the production of asphalt binders that are necessary for the execution of the asphalt coating layers, bond painting, priming and surface treatment; and by the use of fuels for the transportation of materials. As the number of interventions and the thickness of the asphalt coating layer increase, the impact generated in the climate change category also increases.

Other studies have obtained results in which asphalt mixtures generated higher $CO_2$ emissions than those of concrete pavements, including emissions of $NO_x$, $SO_2$ and $CH_4$ [33]. However, when only analyzing the implementation of the structure, without considering long periods, it was found that the asphalt coating is 44% less impactful than the concrete pavement, justified by the manufacture of cement [42]. In this study, an initial difference of 19% was found between the two approaches, with the asphalt being more advantageous. However, at the end of the 30 years, the scenario reverses, with an advantage of 22% for the concrete pavement.

Eutrophication is the phenomenon of the excessive proliferation and growth of aquatic plants that cause interference in a water body. In the implementation of restoration, it has been identified that the recycling approach produces a higher eutrophication impact, as it consumes more aggregates than the other approaches do, in addition to the higher

consumption of diesel by the machinery, in which the machines are in operation for longer due to the layers that are executed. However, when considering the 30-year cycle, the structural reinforcement approach stands out, with a larger number of interventions, higher aggregate consumption and more use of equipment for execution compared to those of the other approaches.

Toxicity is related to the harmful effects of chemicals on human health, whereas ecotoxicity measures the effects on ecosystems. The reinforcement approach is the most harmful, due to the high consumption of asphalt binders. The categories of freshwater ecotoxicity result in lower impacts than the categories of marine ecotoxicity, due to the exploration of oil for the production of fuels and asphalt binders. Human toxicity shows a higher impact than ecotoxicity, as it is influenced not only by the production of asphalt binders but also by the combustion of fuels, especially in the use of equipment for the execution of the work.

Photochemical oxides are secondary pollutants resulting from the burning of fuels and contribute to the appearance of "smog" (smoke or fog). Highly related to the transportation sector, with the contribution of asphalt binder production, aggregate production and diesel consumption, it presents higher impact generation when adopting the structural reinforcement approach, followed by HiMA, recycling and finally Whitetopping, showing that rigid or semi-rigid pavements are more sustainable in the long term.

By analyzing the three case studies and the proposed restoration options, the results show the need to evaluate performance and environmental impacts generated over longer periods, such as the used 30-year period, as adopting a solution only based on its implementation is not consistent. This study provides parameters for generating environmental impacts for the used restoration techniques, being applied in pavement management systems.

With technological advancements and the use of other restoration and maintenance techniques, it is always necessary to seek to evaluate new possibilities, mainly focused on the reuse and recycling of the materials present on the road, reducing the extraction of raw materials, the transportation of materials and waste and the generation of solid waste into the environment.

## 5. Conclusions

Considering that decision making is essential for the good performance of pavement management systems, using tools to evaluate pavement performance from a structural, durability and environmental impact perspective makes results over time more efficient.

By evaluating the studied highways and the addressed restoration techniques, it was found that, for the execution of the structural reinforcement technique, 60% more diesel and fuels are consumed compared to deep recycling over 30 years. When comparing the Whitetopping intervention methodology, for a high-traffic road, the use of this technique becomes advantageous with much lower consumption (6 times less diesel, 11 times less energy and 32 times less fuel oil than those of reinforcement).

In evaluating the impacts generated by the initial restorations, it was identified that, for highways with medium traffic with a pavement thickness of around 5 cm, the restoration technique that has the greatest environmental impact is recycling. However, by increasing the reinforcement thickness to 10 cm, maintaining the characteristics of recycling, reinforcement becomes more damaging in the impact categories related to ecotoxicity, precisely because of the amount of used binder. For highways with high traffic and asphalt mixture thicknesses of around 15 cm, the reinforcement technique is the most impactful regarding ecotoxicity and human toxicity. Recycling stands out for eutrophication, and Whitetopping significantly contributes to acidification and climate change.

However, when evaluating the total impacts generated by each approach over the 30-year period of analysis, it was found that reinforcement has the greatest impact. For SC355 J/BR153, recycling results in approximately 49% less impact than reinforcement. For SC114 P/SJ, recycling reduces the total impact by 46%, and HiMA reduces it by 23%. Finally,

for SC114 L/OC, recycling allows a reduction in emissions of 46%, and Whitetopping has 87% less impact than reinforcement. It is evident that the use of rigid structures such as Whitetopping or semi-rigid structures such as deep recycling with Portland cement generates less environmental impact compared to the use of flexible structures that consume a high amount of asphalt binder and that need interventions at shorter time intervals.

For the boundaries adopted by this study and the 30-year cycle time, it can be concluded that highways with medium traffic have equal impacts in the application of the recycling approach and a difference of approximately 5% in the reinforcement approach, possibly due to the initial characteristics of the degraded pavement. However, when comparing a high-traffic road with a medium-traffic one, the first generates approximately twice the environmental impact both for the reinforcement approach and recycling, whereas the Whitetopping approach has the least impact.

Comparing the consumption and impact approach over a 30-year analysis period, it can be concluded that the initial recommended approach for medium-traffic roads is deep recycling with the addition of Portland cement, with the secondary approach (depending on the structural characteristics of the deteriorated pavement) being the continuous milling approach with reconstruction with SAMI and HiMA. For roads with high vehicle traffic, especially cargo, the Whitetopping technique is the most feasible. However, one must pay attention to the initial characteristics of the road, the imposed traffic, the available materials and equipment and the skilled labor force for execution.

**Author Contributions:** G.L., conceptualization, methodology, software, validation, investigation, resources, data curation, writing—original draft preparation, writing—review and editing, project administration. G.T., formal analysis, visualization, supervision. All authors have read and agreed to the published version of the manuscript.

**Funding:** This research received no external funding.

**Institutional Review Board Statement:** Not applicable.

**Informed Consent Statement:** Not applicable.

**Data Availability Statement:** The original contributions presented in this study are included in the article; further inquiries can be directed to the corresponding author.

**Conflicts of Interest:** The authors declare no conflict of interest.

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
