# Peer review of "Case Study on Life Cycle Assessment Applied to Road Restoration Methods"

_sustainability, doi:10.3390/su15086679_

Round 1

Reviewer 1 Report

It is undoubtedly an interesting and demanding research study. 

Author Response

Dear Reviewer,

The authors would like to thank you for the important comments and suggestions for improving the original manuscript.

As requested, after the manuscript review, it was written a cover letter - to explain, point by point, the details of the revisions and our responses to the referees’ comments.

Please read your considerations followed by our comments below. 

"Point 1: It is undoubtedly an interesting and demanding research study.

 Response 1: Thank you."

For read the manuscript with all corrections, please see the attachment.

Best regards,

The Authors.

Reviewer 2 Report

This paper presents an interesting study which evaluate environmental effects of different restoration and maintenance techniques using the life cycle assessment methodology Analysis Cycle. 

The originality of this study is that the authors link the use of HDM-4 software with the annual analysis of maintenance along with environmental impact studies obtained from the life cycle analysis.

The results are important but they are not easy readable. Could the authors find another way to represent and discuss them?

There are minor issues to be addressed:

1. Check all the text: some sentences seem not to be complete (for example lanes: 91-93, 113-114,  130-132, 265-268
2. Could the authors explain how OpenLCA software works?

3. figure 4: Eight characteristics appear in the histogram while only six appear in the legend. Could the authors explain why?

Author Response

Dear Reviewer,

The authors would like to thank you for the important comments and suggestions for improving the original manuscript.

As requested, after the manuscript review, it was written a cover letter - to explain, point by point, the details of the revisions and our responses to the referees’ comments.

Please read your considerations followed by our comments below. 

"Point 1: This paper presents an interesting study which evaluate environmental effects of different restoration and maintenance techniques using the life cycle assessment methodology Analysis Cycle.

The originality of this study is that the authors link the use of HDM-4 software with the annual analysis of maintenance along with environmental impact studies obtained from the life cycle analysis.

 Response 1: Thank you.

Point 2: The results are important but they are not easy readable. Could the authors find another way to represent and discuss them?

 Response 2: The results and discussions chapters were divided into subchapters:

3. Results

3.1. Initial restorations

3.2. Road SC355 J/BR153

3.2.1. Structural Reinforcement

3.2.2. Deep Recycling

3.3. Road SC114 P/SJ

3.3.1. Structural Reinforcement

3.3.2. Deep Recycling

3.3.3. HiMA

3.4. Road SC114 L/OC

3.4.1. Structural Reinforcement

3.4.2. Deep Recycling

3.4.3. Structural Reinforcement

4. Discussion

4.1. Initial restorations

4.2. Interventions over the 30 years of analysis

Point 3: There are minor issues to be addressed:

1. Check all the text: some sentences seem not to be complete (for example lines: 91-93, 113-114, 130-132, 265-268)

 Response 3:

Lines 108-114 (lines 91-93): When comparing different bituminous mixtures containing recycled materials, namely crumb rubber (CR) and recovered asphalt pavements (RAP), through the results of a life cycle assessment (LCA), it was identified that the use of CR in the pro-duction of asphalt mixtures showed a reduction in the need for gross energy ratio (GER) and Greenhouse Gases (GHG) emissions [46, 47], reduction in noise, especially when the dry process is used [48], improvement in cushioning properties when CR is mixed as an aggregate as a result of vibration absorption by rubber [49].

Lines 128-137 (lines 113-114): The analysis period (life cycle) affects the results and policies adopted for maintenance of the works. In Germany, for example, secondary waste was studied as a substitute for primary materials over a period of 100 years [52]; in China, concrete pavements with a life cycle of 20 and 40 years [53]; in Korea, different civil structures were compared over 50 years [54]; in Spain, sustainable maintenance program methods of pavements under budget constraints were analyzed, considering periods of 25 years, however, there are studies developed that contemplate analyses between 15 and 50 years [55]; and in the United States of America, one of the countries that develops the most studies in the LCA area, the life cycle of asphalt pavement recycling was evaluated with respect to GHG emissions with a temporal aspect of 40 and 100 years [56].

Lines 148-151 (lines 130-132): The OpenLCA [59] software was used for environmental modeling, followed by the presentation and analysis of the results of the environmental impacts, generated by each type of intervention adopted for each impact category, providing parameters to support decision-making in the management of road.

Lines 279-285 (lines 265-268): Stage 1 - Production of raw materials and mixtures: in this stage, all inputs and all outputs of the system (inventory obtained from simulation in the HDM-4 software) are listed, including the extraction and crushing of aggregates, the production of bind-ers (Portland cement, asphalt cement, cutback asphalt, asphalt emulsion), and the milling of mixtures (graded aggregate, asphalt mixtures, Portland cement concrete), in addition to quantifying the transport of inputs between the extraction site, the place of production, and the place application, as well as the activities and equipment neces-sary for these services to be successfully completed.

Point 4: 2. Could the authors explain how OpenLCA software works?

Response 4: OpenLCA is open source software for Life Cycle Assessment (LCA) and sustainability assessment. It has been developed by GreenDelta since 2006 (www.greendelta.com). As open source software, it is freely available with no license fees (www.openlca.org). The source code can be viewed and changed by anyone. Furthermore, the open source nature of the software makes it very suitable for use with sensitive data. The software, as well as any model created, can be shared freely, as long as the database license allows it.

OpenLCA supports the following import formats: zolca, Ecospold1, Ecospold2, Excel, ILCD, SimaPro CSV.

The data quality system can be selected from the available systems in the “Data quality systems” directory in the “Indicators and parameters” section of the active database.

An LCIA openLCA method package is available at www.openlca.org/downloads. This comprehensive package of environmental impact assessment methods is formatted for use with all databases available on the openLCA Nexus, including, for example, ecoinvent 3, GaBi and ELCD. This package includes normalization and weighting, provided this is provided for by the method.

A reference was added to the article that explains how to proceed with the use of the software

Line 148 – 151: The OpenLCA [59] software was used for environmental modeling, followed by the presentation and analysis of the results of the environmental impacts, generated by each type of intervention adopted for each impact category, providing parameters to support decision-making in the management of road.

Line 1066-1069: 59. Dr A. Ciroth, C. Di Noi, T. Lohse, M. Srocka. openLCA 1.10 Comprehensive User Manual. GreenDelta GmbH, Berlin, Germany, 2019. https://www.openlca.org/wp-content/uploads/2020/01/openLCA_1.10_User-Manual.pdf

Point 5: 3. figure 4: Eight characteristics appear in the histogram while only six appear in the legend. Could the authors explain why?

 Response 5: It was hidden, adjusted. Lines 469-470

For read the manuscript with all corrections, please see the attachment.

Best regards,

The Authors.

Reviewer 3 Report

The manuscript sustainability-2257855 "Case study on life cycle assessment applied to road restoration methods" deals with a very important topic such is the impact of roads maintenance. The paper is interesting and analyze the environmental impacts of several restoration works.

This paper presents excellent basis for future adoptions of  of green strategies to manage roads requires deep knowledge to implement effective procedures at different life cycle stages.

The paper is technically prepared by instructions for authors; the paper presents novel approach in the field.

However, I believe that the manuscript should be improved before published to the journal.

line 48-49: "The reliability and accuracy of an LCA are affected by the reliability of the adopted methodologies and models" what about input data and inventory?

line 50: " range of values" is not true for all LCA analyses.

line 51: to which possibilities are the authors referring?

line 58: about road construction you can consider: "Trunzo, G., Moretti, L., & D'Andrea, A. (2019). Life cycle analysis of road construction and use. Sustainability (Switzerland), 11(2) doi:10.3390/su11020377" and "Moretti, L. (2022). How road cross-sections affect the environmental impacts from cradle to grave. Cleaner Environmental Systems, 6 doi:10.1016/j.cesys.2022.100088"

line 78: standardization in life cycle is provided in Europe by the standard EN 15804:2019 for building materials.

please check all "2" of CO2. The same for CH4, N2O and all chemical formulas.

please revise the sentence at lines 91-93. I did not find the verb.

line 42 vs 122: LCA is life cycle analysis or life cycle assessment?

line 451: I did not find the normalization process. Please explain the procedure.

Author Response

Dear Reviewer,

The authors would like to thank you for the important comments and suggestions for improving the original manuscript.

As requested, after the manuscript review, it was written a cover letter - to explain, point by point, the details of the revisions and our responses to the referees’ comments.

Please read your considerations followed by our comments below. 

Point 1: The manuscript sustainability-2257855 "Case study on life cycle assessment applied to road restoration methods" deals with a very important topic such is the impact of roads maintenance. The paper is interesting and analyze the environmental impacts of several restoration works.

This paper presents excellent basis for future adoptions of  of green strategies to manage roads requires deep knowledge to implement effective procedures at different life cycle stages.

The paper is technically prepared by instructions for authors; the paper presents novel approach in the field.

However, I believe that the manuscript should be improved before published to the journal.

 Response 1: The authors are grateful for the considerations and present below the changes with their respective justifications.

Point 2: line 48-49: "The reliability and accuracy of an LCA are affected by the reliability of the adopted methodologies and models" what about input data and inventory?

Response 2: Adjusted the text of lines 51-54: The reliability and accuracy of an LCA are affected by the reliability of the adopt-ed methodologies and models. Moreover, these models require the estimation of input parameters, inventory development and methodological choices, such as that may impact the results significantly [7].

Point 3: line 50: "range of values" is not true for all LCA analyses.

line 51: to which possibilities are the authors referring?

 Response 3:

The authors chose to remove the questioned phrase, due to the use of the term "single figure", as they understand that there are countless interferences in environmental modeling. However, it lists the citations where the author of the thesis mentions the content:

In abstract: ... These research outcomes highlight the importance of incorporating uncertainty into pavement LCA. The reliability and accuracy of an LCA is affected by the reliability of the methodologies and models adopted. LCA results should not be presented as ’single figure’ absolute values, but rather considering a range of values to reflect the uncertainties and variability that lie behind them.

In chapter 7: These research outcomes highlight the importance of incorporating uncertainty into pavement LCA. The reliability and accuracy of an LCA is affected by the reliability of the assumptions, methodologies and models adopted. LCA results should not be presented as ’single figure’ absolute values, but rather as a range of values to estimate the uncertainties and variability that lie behind them.

Point 4: line 58: about road construction you can consider: "Trunzo, G., Moretti, L., & D'Andrea, A. (2019). Life cycle analysis of road construction and use. Sustainability (Switzerland), 11(2) doi:10.3390/su11020377" and "Moretti, L. (2022). How road cross-sections affect the environmental impacts from cradle to grave. Cleaner Environmental Systems, 6 doi:10.1016/j.cesys.2022.100088"

 Response 4: Two citations were included, as recommended.

Lines 69-77: In order to choose the process with the least impact, several environmental im-pact indicators must be considered, since the contribution of each step related to the LCA differs significantly between each analyzed parameter. High volumes of traffic combined with the geometric layout present variation in the degree of environmental sustainability. Therefore, social and economic criteria must be integrated in the com-prehensive evaluation of road works [15]. Considering the LCA from construction to use, it was identified that roads that do not need tunnels and bridges have lower envi-ronmental impacts. Flexible pavements present lower initial impacts when compared to rigid ones, however, when considering prolonged periods of analysis, the behavior is reversed [16].

Point 5: line 78: standardization in life cycle is provided in Europe by the standard EN 15804:2019 for building materials.

Response 5: Include text in lines 96-98: In Europe, EN 15804:2012+A2:2019 is used to conduct environmental studies, through the Life Cycle Assessment of construction products and services, however it does not include social and economic studies [41].

Point 6: please check all "2" of CO2. The same for CH4, N2O and all chemical formulas.

Response 6: The text was revised.

Point 7: please revise the sentence at lines 91-93. I did not find the verb.

Response 7: Lines 108-114: When comparing different bituminous mixtures containing recycled materials, namely crumb rubber (CR) and recovered asphalt pavements (RAP), through the re-sults of a life cycle assessment (LCA), it was identified that the use of CR in the pro-duction of asphalt mixtures showed a reduction in the need for gross energy ratio (GER) and Greenhouse Gases (GHG) emissions [46, 47], reduction in noise, especially when the dry process is used [48], improvement in cushioning properties when CR is mixed as an aggregate as a result of vibration absorption by rubber [49].

Point 8: line 42 vs 122: LCA is life cycle analysis or life cycle assessment?

Response 8: LCA is life cycle assessment, adjusted the expression in line 42.

Line 46: The Life Cycle Assessment (LCA) methodology is a useful tool for achieving a complete overview of a product or process. …

Line 140: This study aims to combine the use of the HDM-4 software with the annual analysis of maintenance and restorations together with studies of the environmental im-pacts obtained in the life cycle assessment (LCA), contributing technically and scientifically,…

Point 9: line 451: I did not find the normalization process. Please explain the procedure.

Response 9: Normalization was not applied, but relative impacts. The chart shows the relative indicator results of the respective project variants. For each indicator, the maximum result is set to 100% and the results of the other variants are displayed in relation to this result. Therefore, the used term was adjusted.

For read the manuscript with all corrections, please see the attachment.

Best regards,

The Authors.

Reviewer 4 Report

Dear Author,

Thank you for the good read. I have listed the comment below.

General comment

1)    The paper has lots of data and is well-studied along with field implementation.

2)    The Result and discussion are a bit missed up and could get better if it is clearly written.

3)    Under the section on results and especially discussion, inserting a subsection would make it easier to understand and to keep track. The subsection may be by grouping the similar environmental impact or based on different interventions/restoration.

4)    Title: “Case study on life cycle assessment applied to road restoration methods” could be modified to express the methods/best method. It will be easier for readers if they can understand the essence of the paper through the topic.

5)    Tables are not in format and not uniform

6)    Some of the letters in the figure are not readable eg. Figure 1

7)    The objective/novelty of the paper could be clearly highlighted. It doesn’t mention clearly what the authors want to find out.

8)    All references/citations before 2017-2018. This paper could use some newly updated references.

9)    It is encouraged to use references/citations in the discussion section to strengthen your findings.

10) The paper could do some language checking.

Line wise comments

1)    Line 9: the word “investments” is used twice which can be avoided.

2)    Line 13-24: In the abstract try to use the full form instead of the abbreviations.

3)    Include a sentence on the methodology in the abstract.

4)    Line 23: Keyword may include the word related to restorations too.

5)    Line 30, 53,106,107, 303 and others: CO2 must be written with “2” as a subscript. Please check throughout the paper for this.

6)    Line 38-41: the sentence is not clear.

7)    Line 43-45: This citation is not proper: The author has cited the paper “Santero, N. J.; Masanet, E.; Horvath, A. Life-cycle assessment of pavements. Part I: Critical review. Resources, Conservation and Recycling, Vol. 55, p. 801–809, 2011” here but the paper didn’t mention the statement in the context. Please revisit the paper.

8)    Line 52: variability or variabilities??

9)    Line 61-62: this sentence could be rephrased for better understanding

10) Line 62: there are 2 sentences but no “full stop”

11) Line 69-71: I could understand the sentence. Please revisit.

12) Line 72: “life cycle impact assessment” can be written as LCIA since u have defined it in the previous section.

13) Line 91-93: is it an incomplete sentence?

14) Line 94: “GHG” emissions. Expand the abbreviation here since it's written for the first time on paper.

15) Line 93-97: can you restructure the sentence?

16) Line 127-133: Can you restructure/ rephrase the sentence for a clear understanding

17) Line 138-139: the sentence looks incomplete

18) Line 261-262: repetition of word input and output. Does this have significance?

19) Line 349: life cycle inventory (LCI)- use abbreviation since it’s defined above.

20) Line 705/: chart 1 & 2: What are RE, RC HM and WH in the table?

21) Line 757-758: it needs citation if it’s not the paper's finding.

Author Response

Dear Reviewer,

The authors would like to thank you for the important comments and suggestions for improving the original manuscript.

As requested, after the manuscript review, it was written a cover letter - to explain, point by point, the details of the revisions and our responses to the referees’ comments.

Please read your considerations followed by our comments below. 

Point 1: Thank you for the good read. I have listed the comment below.

 Response 1: Thank you.

Point 2: General comment

1) The paper has lots of data and is well-studied along with field implementation.

2) The Result and discussion are a bit missed up and could get better if it is clearly written.

3) Under the section on results and especially discussion, inserting a subsection would make it easier to understand and to keep track. The subsection may be by grouping the similar environmental impact or based on different interventions/restoration.

 Response 2: The results and discussions chapters were divided into subchapters:

3. Results

3.1. Initial restorations

3.2. Road SC355 J/BR153

3.2.1. Structural Reinforcement

3.2.2. Deep Recycling

3.3. Road SC114 P/SJ

3.3.1. Structural Reinforcement

3.3.2. Deep Recycling

3.3.3. HiMA

3.4. Road SC114 L/OC

3.4.1. Structural Reinforcement

3.4.2. Deep Recycling

3.4.3. Structural Reinforcement

4. Discussion

4.1. Initial restorations

4.2. Interventions over the 30 years of analysis

Point 3: 4) Title: “Case study on life cycle assessment applied to road restoration methods” could be modified to express the methods/best method. It will be easier for readers if they can understand the essence of the paper through the topic.

Response 3: Title changed to: Case Study on Life Cycle Assessment Applied to Different Pavement Restoration Alternatives

Point 4: 5) Tables are not in format and not uniform.

Response 4: Charts 1 and 2 were changed to tables.

Following the provided template, two forms of table were adopted, due to the amount of data presented in the last two tables.

Point 5: 6) Some of the letters in the figure are not readable eg. Figure 1

Response 5: The font size of the text in the figure has been adjusted to be visible.

Point 1: 7) The objective/novelty of the paper could be clearly highlighted. It doesn’t mention clearly what the authors want to find out.

Response 7: The authors adjusted the text between lines 138-141 in order to make the objective of the article clear: This study aims to combine the use of the HDM-4 software with the annual analy-sis of maintenance and restorations together with studies of the environmental im-pacts obtained in the life cycle assessment (LCA), contributing technically and scien-tifically to decision-making in choosing the appropriate alternative.

Point 8: 8) All references/citations before 2017-2018. This paper could use some newly updated references.

Response 8: Two recent works have been added

Trunzo, G., Moretti, L., & D'Andrea, A. Life cycle analysis of road construction and use. Sustainability (Switzerland), 11(2), 377, 2019. https://doi.org/10.3390/su11020377

Moretti, L. How road cross-sections affect the environmental impacts from cradle to grave. Cleaner Environmental Systems, 6, 100088, 2022. https://doi.org/10.1016/j.cesys.2022.100088

Lines 69-77: In order to choose the process with the least impact, several environmental im-pact indicators must be considered, since the contribution of each step related to the LCA differs significantly between each analyzed parameter. High volumes of traffic combined with the geometric layout present variation in the degree of environmental sustainability. Therefore, social and economic criteria must be integrated in the com-prehensive evaluation of road works [15]. Considering the LCA from construction to use, it was identified that roads that do not need tunnels and bridges have lower envi-ronmental impacts. Flexible pavements present lower initial impacts when compared to rigid ones, however, when considering prolonged periods of analysis, the behavior is reversed [16].

Point 9: 9) It is encouraged to use references/citations in the discussion section to strengthen your findings.

Response 9: Thank you.

Point 10: 10)The paper could do some language checking.

Response 10: The text underwent language revision.

Point 11: Line wise comments

1) Line 9: the word “investments” is used twice which can be avoided.

Response 11: Abstract: Brazil's dependence on road transportation, combined with the high extent of the network and the lack of investment management in maintenance and restoration, make traffic conditions poor, resulting in unwanted costs and environmental impacts.

Point 12: 2) Line 13-24: In the abstract try to use the full form instead of the abbreviations.

Response 12: … in the HDM-4 (Highway Development and Management) software…

Point 13: 3) Include a sentence on the methodology in the abstract.

Response 13: Lines 13-17: To develop the study, the ecoinvent database and the OpenLCA software were used to model, based on studies developed in the HDM-4 (Highway Development and Management) software, the interventions applied in the initial year and in 30 years. Using the life cycle assessment methodology, the environmental impacts generated for the categories acidification, climate change, eutrophica-tion, ecotoxicity, human toxicity and photochemical oxidation were identified.

Point 14: 4) Line 23: Keyword may include the word related to restorations too.

Response 14: The following words were included in the keywords: Whitetopping, reinforcement, recycling and HiMA.

Line 25-26: Keywords: Life cycle assessment; environmental impact; restoration; roads; whitetopping; reinforcement; recycling; HiMA.

Point 15: 5) Line 30, 53,106,107, 303 and others: CO2 must be written with “2” as a subscript. Please check throughout the paper for this..

Response 15: The text was revised.

Point 16: 6) Line 38-41: the sentence is not clear.

Response 16: Adjusted the text of lines 41-45: The use of environmental approaches for decision-making in construction projects is becoming more common, implemented through waste management actions, the con-trol of the exploitation of natural resources and the reduction of noise, gaseous, liquid or solid pollution [3].

Point 17: 7) Line 43-45: This citation is not proper: The author has cited the paper “Santero, N. J.; Masanet, E.; Horvath, A. Life-cycle assessment of pavements. Part I: Critical review. Resources, Conservation and Recycling, Vol. 55, p. 801–809, 2011” here but the paper didn’t mention the statement in the context. Please revisit the paper.

Response 17: Adjusted text, lines 46-50: The Life Cycle Assessment (LCA) methodology is a useful tool for achieving a complete overview of a product or process. LCA is an essential process for achieving conclusions and maintenance strategies or global projects for the entire lifespan, standardizing functional units, expanding the scope of studies, improving quality and reliability, and expanding boundary systems [6].

Point 18: 8) Line 52: variability or variabilities??

Response 18: The word must be written in the plural, adjusted in the text.

Lines 51-54: The reliability and accuracy of an LCA are affected by the reliability of the adopt-ed methodologies and models. Moreover, these models require the estimation of input parameters, inventory development and methodological choices, such as that may impact the results significantly [7].

Point 19: 9) Line 61-62: this sentence could be rephrased for better understanding

10) Line 62: there are 2 sentences but no “full stop”

Response 19: Adjusted the text of lines 64-698 Most of the road investment is related to the exploration of raw materials, the labor for execution and the transport of materials, which make up most of the maintenance and operation costs [13]. In Brazil, approximately 12.5% of the total budget invested in São Paulo, in 4 years, was applied in the development and maintenance of roads [14].

Point 20: 11) Line 69-71: I could understand the sentence. Please revisit.

Response 20: Adjusted the text of lines 82-86: A meticulous quantification of the environmental impacts of pavements requires in-formation from numerous sources related to the stages of its life cycle, information that is not always available. LCA studies are subject to assumptions and simplifications regarding their scope, system limits and data, inevitably leading to uncertainties in LCA assessments [17].

Point 21: 12) Line 72: “life cycle impact assessment” can be written as LCIA since u have defined it in the previous section.

Response 21: The description "life cycle impact assessment" was replaced by the abbreviation LCIA.

Line 87: The lack of standardization in LCIA has been discussed against temporal standards…

Point 22: 13) Line 91-93: is it an incomplete sentence?

14) Line 94: “GHG” emissions. Expand the abbreviation here since it's written for the first time on paper.

15) Line 93-97: can you restructure the sentence?

Response 22: Expanded abbreviation as recommended by the reviewer and adjusted the text of lines 108-114: When comparing different bituminous mixtures containing recycled materials, namely crumb rubber (CR) and recovered asphalt pavements (RAP), through the re-sults of a life cycle assessment (LCA), it was identified that the use of CR in the pro-duction of asphalt mixtures showed a reduction in the need for gross energy ratio (GER) and Greenhouse Gases (GHG) emissions [46, 47], reduction in noise, especially when the dry process is used [48], improvement in cushioning properties when CR is mixed as an aggregate as a result of vibration absorption by rubber [49].

Point 23: 16) Line 127-133: Can you restructure/ rephrase the sentence for a clear understanding.

Response 23: Adjusted the text of lines 145-151: Initially, case studies are presented with their respective characteristics, the results obtained in the HDM-4 software with the frequency of interventions, followed by the quantification of consumption for all stages from raw material extraction to the end of the analysis cycle, obtaining the inventory. The OpenLCA [59] software was used for environmental modeling, followed by the presentation and analysis of the results of the environmental impacts, generated by each type of intervention adopted for each impact category, providing parameters to support decision-making in the man-agement of road.

Point 24: 17) Line 138-139: the sentence looks incomplete

Response 24: Adjusted the text of lines 156-160: These constructions were chosen because they present the structures and methodologies of restorations targeted for the study and because we have knowledge of the characteristics of the materials (laboratory tests), the executive processes used and the evaluation of post construction performance.

Point 25: 18) Line 261-262: repetition of word input and output. Does this have significance?

Response 25: Adjusted text, excluding words that were repeated between parentheses.

Point 26: 19) Line 349: life cycle inventory (LCI)- use abbreviation since it’s defined above.

Response 26: Changed as per reviewer's recommendation..

Point 27: 20) Line 705/: chart 1 & 2: What are RE, RC HM and WH in the table?

Response 27: A note has been added at the end of the table to describe the meaning of each abbreviation used in the table body. RE (Reinforcement), RC (Recycling), HM (HiMA), WH (Whitetopping).

Point 28: 21) Line 757-758: it needs citation if it’s not the paper's finding.

Response 28: This text is a discovery of the research, therefore it remained without citation.

For read the manuscript with all corrections, please see the attachment.

Best regards,

The Authors.

Reviewer 5 Report

 This study evaluated the environmental performance of three roads, applying different restoration and maintenance techniques throughout the analysis cycle. The life cycle assessment methodology was used to analyze environmental impacts. The research idea of the paper is clear, and the data is detailed. It is recommended to accept the paper. 

Author Response

Dear Reviewer,

The authors would like to thank you for the important comments and suggestions for improving the original manuscript.

As requested, after the manuscript review, it was written a cover letter - to explain, point by point, the details of the revisions and our responses to the referees’ comments.

Please read your considerations followed by our comments below. 

Point 1: This study evaluated the environmental performance of three roads, applying different restoration and maintenance techniques throughout the analysis cycle. The life cycle assessment methodology was used to analyze environmental impacts. The research idea of the paper is clear, and the data is detailed. It is recommended to accept the paper.

Response 1: Thanks for the considerations and recommendation for acceptance.

For read the manuscript with all corrections, please see the attachment.

Best regards,

The Authors.

Round 2

Reviewer 2 Report

No comments

Reviewer 4 Report

Dear Authors,

Thank you for considering all the comments provided in the previous review. It looks good to me.

Good luck